# Study on underwater noise characteristics and mechanism of discharge flow from wide-crested weir

Qingxiang Shui[1], Daguo Wang[2]*, Yuxin Chen[2], Tao Yu[2], Yi Fan[2], Yang Dai[2]

1  Ministry of Education, Key Laboratory of Fluid and Power Machinery(Xihua University), Chengdu, China,
2  School of Civil Engineering and Geomatics, Southwest Petroleum University, Chengdu, China

* dan_wangguo@163.com

## Abstract

To provide theoretical guidance and technical support for mitigating underwater noise generated by discharge flow from wide-crested weirs, this study integrates physical experiments and numerical simulations. The effects of inflow, downstream static water depth, and weir height on underwater noise sound pressure level (SPL) and fluctuating pressure SPL were analyzed using correlation methods, and the time–frequency variation mechanism of underwater noise was investigated through wavelet analysis. Results show a significant positive correlation between underwater noise SPL and fluctuating pressure SPL, indicating that fluctuating pressure can be used to characterize underwater noise variation. Downstream static water depth exerts the greatest influence, with a negative correlation to SPL, whereas inflow and weir height have relatively smaller effects, both showing positive correlations. As inflow and weir height increase, and static water depth decreases, the mid-frequency range (400~600 Hz) exhibits high energy that decays rapidly. Fluctuating pressure at the measuring point is affected by the impingement of the main flow tongue on the downstream water body, vortex structure variation and breakup, and sound waves radiated from bubble collapse.

## 0  Introduction

Wide-crested weirs are common water conservancy landscape facilities [1]. During discharge, gas entrainment and liquid splashing occur, producing a mixed flow of water and bubbles with a free surface [2]. This process includes bubble formation, coalescence, and rupture, as well as interactions with the solid wall surface. These complex phenomena lead to underwater noise [3]. Urban noise pollution has become one of the most serious environmental concerns for residents, and underwater noise generated by discharge from wide-crested weirs exacerbates this problem, affecting the physical and mental health of nearby populations [4,5]. Furthermore, free-surface bubble flow is also typical during high-dam discharge. The associated underwater

**Data availability statement:** All relevant data are available in the Mendeley Data repository at:SHUI, QINGXIANG (2026), "Study on underwater noise characteristics and mechanism of discharge flow from wide-crested weir", Mendeley Data, V1, doi: https://doi.org/10.1016/10.17632/knd3x9wcbx.1.

**Funding:** This study was supported in the form of funding by the National Natural Science Foundation of China (Grant Number 42171108) awarded to Daguo Wang, the Sichuan International Science and Technology Innovation Cooperation Project (Grant Number 2025YFHZ0223) to Qingxiang Shui, and the Open Research Subject of Key Laboratory of Fluid and Power Machinery (Xihua University), Ministry of Education (Grant Number LTDL-2025014) to Qingxiang Shui. These funders had player a role in study design, data collection and analysis, decision to publish, and preparation of the manuscript.

**Competing interests:** The authors have declared that no competing interests exist.

noise alters aquatic ecosystems [6], influencing the density, distribution, and behavior of organisms, and may cause hearing damage or even mortality in fish [7].

Beyond ecological effects, the dynamics of underwater noise are of interest in a range of physical and engineering contexts, including cavitation on ship propellers [8], underwater explosions [9], and ultrasonic cleaning [10]. Unlike cavitation-induced noise, bubble-flow noise arises from multiscale aggregation and rupture of entrained air bubbles [11]. In addition, the free surface reflects sound waves and affects their damping [12,13], with noise near the free surface concentrating mainly in the high-frequency range [14].

Current research has focused predominantly on single-phase flow noise, whereas noise generated by free-surface bubble flows has been less studied [15]. The mechanism is highly complex, characterized by bubble morphology evolution and acoustic propagation influenced by bubbles and the free surface [16]. Deane and Stokes [17] developed a theoretical model for noise from breaking waves, validated against laboratory plunging-wave data, and attributed noise from several hundred Hz up to 80 kHz to acoustic pulses radiated by collapsing bubbles. The total level and spectrum depended on bubble formation rate, damping, and acoustical skin depth. Johnson et al. [3] investigated a low-head weir dam, evaluating its acoustic environment and reporting that flow rate and weir head significantly influence discharge noise. Li et al. [18] performed physical model tests on discharge noise from a ridge-free wide-crested weir, analyzing the effects of weir height, downstream static water depth, and flow rate on pressure and noise. Hu [19] characterized noise spectra for wide-crested and WES weirs under overflow conditions. Fuster and Montel [20] demonstrated that bubble initiation, deformation, and coalescence affect flow fluctuations and generate new acoustic sources. Guo et al. [21,22] established relationships between discharge noise, weir type, and flow rate through spectral and fluctuating-pressure analyses. Collectively, these studies show that fluctuating pressure is closely associated with discharge noise and reflects unsteady flow characteristics.

With advances in computational fluid dynamics, numerical simulations provide an effective means to resolve complex flow fields at high spatial and temporal scales. Simulations allow detailed study of fluctuating pressure fields and noise generation mechanisms [23]. Xue et al. [24] and Fan et al. [25] analyzed flow field evolution using numerical models, identifying high-energy frequency bands in fluctuating pressure spectra. Lian et al. [26] showed that shear layers around high-speed submerged jets and wall jets dominate low-frequency noise sources in plunge pools. Shen [27] validated regular wave generation in a three-dimensional flume using OpenFoam. Li et al. [28] applied a standard $k - \varepsilon$ turbulence model with volume-of-fluid (VOF) coupling to analyze overtopping flows and shear stress distribution. Liang [29] demonstrated that water noise is influenced by both turbulence and aeration concentration, deriving a regression between noise level, aeration, and Froude number. Zhang et al. [30] described spatial distributions of bubble and noise characteristics in jet-entrainment flows. These works confirm that numerical simulation can capture the acoustic–flow coupling processes in bubble flows with a free surface.

In this study, physical experiments and numerical simulations are integrated to examine the effects of inflow, downstream static water depth, and weir height on underwater noise sound pressure level (SPL) and fluctuating pressure SPL. Correlation analysis quantifies the relationship between fluctuating pressure and noise. Wavelet analysis is employed to reveal the time–frequency distribution of signals, identify noise sources, and elucidate variation under different hydraulic conditions. The findings provide theoretical guidance for the design of landscape weirs aimed at reducing urban noise pollution, as well as support for mitigating the ecological effects of high-dam discharge.

## 1 Physical experiments

### 1.1 Model layout

Fig 1 shows the longitudinal centerline section of the flume. The flume is 24 m long, 1.2 m wide, and 1.2 m deep. The wide-crested weir is 0.5 m long, 1.2 m wide, and $H$ high, and is located 14 m downstream from the inlet. The $x$ direction corresponds to the flow direction. To enable free-falling discharge flow, vent holes were placed 0.1 m below the rear crest of the weir. Each hole has a diameter of 0.03 m, and 15 holes were evenly distributed along the span. Underwater noise was measured with hydrophone V1, an RHS(A)-30 spherical hydrophone produced by the Hangzhou Institute of Applied Acoustics. Its working frequency range is 20 Hz–50 kHz, and its voltage sensitivity is −197 dB. The hydrophone and pressure probe location was selected based on preliminary tests showing that this position captures the strongest pressure fluctuations associated with jet impingement and bubble entrainment. The hydrophone was positioned at the channel bottom along the longitudinal axis, 0.3 m downstream of the weir. Fluctuating pressure was measured by sensor P1 at a location nearly coincident with the hydrophone, enabling direct comparison between underwater noise and pressure fluctuations. Although measurements were performed at a single downstream location, this point was selected as a representative position for pressure fluctuation and noise generation, enabling consistent comparison across all tested conditions. A USB-1252A multifunction data acquisition card (12-bit resolution, maximum sampling rate 500 kS/s, 16 analog channels) was used to form the pressure measurement system for the bubbly flow.

### 1.2 Experimental setup

During testing, the hydrophone sampling frequency was set at 32 kHz, with an acquisition time of 60 s. Acoustic signals were Fourier transformed, then divided into 1/3 octave bands, yielding 29 underwater noise bands in the range 20 Hz–12,500 Hz. To improve data reliability and reduce random error, underwater noise and fluctuating pressure were measured simultaneously. For each condition, three acquisitions were taken, and the mean of the three total SPLs was used [22].

Experiments covered five inflow discharges ($Q$ = 15, 30, 45, 60, and 75 L/s), five downstream static water depths ($h$ = 10, 20, 30, 40, and 50 cm), and five weir heights ($H$ = 50, 60, 70, 80, and 90 cm). In total, 120 flow-over conditions were analyzed.

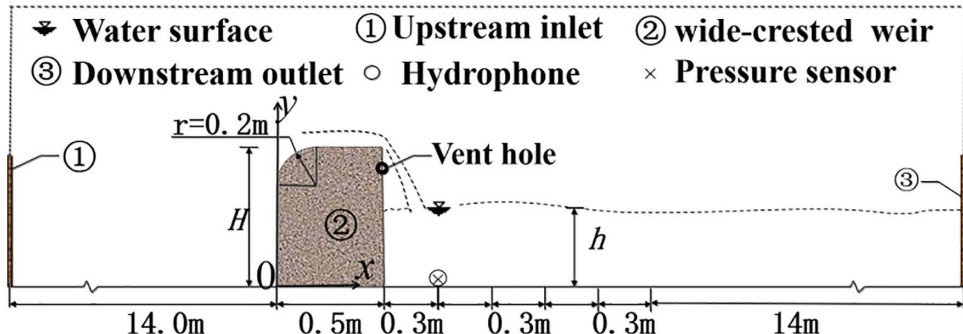

**Fig 1. Section view of the longitudinal central axis of the flume.**

## 1.3 Data processing

Two pressure-based quantities are analyzed: underwater noise sound pressure level (SPL) and fluctuating pressure SPL. These quantities are standard acoustic indicators widely used to characterize respectively the overall noise energy and the intensity of unsteady pressure fluctuations.

**1.3.1 Underwater noise data processing.** Raw hydroacoustic pressure signals were processed using the fast fourier transform (FFT). The procedure for obtaining unweighted total SPL in 1/3 octave bands followed [31]:

1) Import pressure signal and perform FFT (sampling rate 32 kHz).

2) Apply a Hanning window to obtain the preliminary spectrum.

3) Divide the frequency spectrum into 1/3 octave bands.

Use the vibration velocity level method to convert into unweighted SPL in each band. This yielded 29 frequency-band SPLs between 20 Hz and 12,500 Hz. The unweighted SPLs across all bands were summed to obtain the total underwater noise level:

$$L_{pw} = 10\lg\sum_{i=1}^{n} 10^{0.1L_{pi}}$$

(1)

where $L_{pw}$ is the total underwater noise SPL, $L_{pi}$ is the unweighted SPL of the i-th band, and frequency band number $n = 29$.

**1.3.2 Data processing of fluctuating pressure SPL.** The fluctuating pressure SPL represents the ratio of the root mean square (RMS) value of fluctuating pressure $P_{rms}$ to the reference sound pressure $P_{ref}$. It is calculated as follows [32–33]:

$$\overline{P} = \frac{1}{T}\int_0^T P(t)\,dt$$

(2)

$$P_{rms} = \sqrt{\lim_{T\to\infty}\frac{1}{T}\int_0^T \left(P(t) - \overline{P}\right)^2}$$

(3)

$$SPL = 20\lg\left[\frac{P_{rms}}{P_{ref}}\right]$$

(4)

where $T$ is the statistical time of measuring pressure, $P(t)$ is the statistical value of pressure, $\overline{P}$ is the mean value of pressure, and the reference sound pressure in water is $P_{ref} = 1 \times 10^9$ kPa.

## 2 Numerical methods

### 2.1 Governing equations and numerical methods

**2.1.1 Governing equations.** The standard $k$–$\varepsilon$ turbulence model is applied to simulate three-dimension unsteady incompressible flow without heat conduction. The continuity and momentum equations are expressed as:

$$\frac{\partial \overline{U}_i}{\partial x_i} = 0$$

(5)

$$\frac{\partial \overline{U}_i}{\partial t} + \overline{U}_j \frac{\partial k}{\partial x_j} = -\frac{1}{\rho}\frac{\partial p}{\partial x_i} + \frac{\partial}{\partial x_j}\left(\mu \frac{\partial \overline{U}_i}{\partial x_j} - \overline{u_i u_j}\right) + \boldsymbol{F}_s \tag{6}$$

$$\overline{u_i u_j} = \frac{2}{3} k \delta_{ij} - \mu_t \left(\frac{\overline{U}_i}{\partial x_j} + \frac{\overline{U}_j}{\partial x_i}\right), \ \mu_t = C_\mu \frac{k^2}{\varepsilon} \tag{7}$$

where $\rho$ and $\mu$ represent the density and viscosity of the computational cell, respectively; $t$ and $\overline{p}$ denote time and pressure; $\mu_t$ is the turbulent eddy viscosity; $k$ is the turbulent kinetic energy; $\varepsilon$ is the turbulent dissipation rate, with $C_\mu = 0.09$; $\boldsymbol{F}_s$ is the surface tension source term, given by:

$$\boldsymbol{F}_s = \sigma \kappa \boldsymbol{n} \tag{8}$$

where $\sigma$ represents the surface tension coefficient; $\boldsymbol{n} = \nabla \alpha$ is the normal vector to the free surface; $\kappa = -\nabla \cdot \boldsymbol{n}$ denotes the average curvature of the free interface.

The transport equations for $k$ and ε in the standard $k$–ε model are as follows [34]:

$$\frac{\partial k}{\partial t} + \overline{U}_j \frac{\partial k}{\partial x_j} = \frac{\partial}{\partial x_j}\left(\frac{\mu_t}{\sigma_k}\frac{\partial k}{\partial x_j}\right) + \mu_t \left(\frac{\overline{U}_i}{\partial x_j} + \frac{\overline{U}_j}{\partial x_i}\right)\frac{\partial \overline{U}_j}{\partial x_j} - \varepsilon \tag{9}$$

$$\frac{\partial \varepsilon}{\partial t} + \overline{U}_j \frac{\partial \varepsilon}{\partial x_j} = \frac{\partial}{\partial x_j}\left(\frac{\mu_t}{\sigma_\varepsilon}\frac{\partial \varepsilon}{\partial x_j}\right) + \frac{\varepsilon}{k}\left(C_{1\varepsilon}\mu_t \left(\frac{\overline{U}_i}{\partial x_j} + \frac{\overline{U}_j}{\partial x_i}\right)\frac{\partial \overline{U}_j}{\partial x_j} - C_{2\varepsilon}\varepsilon\right) \tag{10}$$

In the equation, $C_{1\varepsilon}$ and $C_{2\varepsilon}$ are empirical constants; $\sigma_k$ and $\sigma_\varepsilon$ are the turbulent Prandtl numbers for $k$ and ε, respectively.

**2.1.2 Numerical discretization and computational methods.** The interFoam solver in OpenFOAM is used to simulate two-phase water–air flow, based on the Volume of Fluid (VOF) method [35–36]. In VOF, a scalar field α represents the phase fraction of the fluid in each computational cell. The mixture properties (viscosity and density) are determined by volume-weighted averages:

$$\mu = (1 - \alpha)\mu_g + \alpha \mu_l \tag{11}$$

$$\rho = (1 - \alpha)\rho_g + \alpha \rho_l \tag{12}$$

where $\alpha$ denotes the liquid-phase volume fraction, and the subscripts $l$ and $g$ refer to liquid and gas, respectively. The α field is defined as:

The phase fraction is defined as:

$$\alpha = \begin{cases} 1 & \text{(liquid)} \\ 0 < \alpha < 1 & \text{(gas-liquid interface)} \\ 0 & \text{(gas)} \end{cases} \tag{13}$$

The value range of $\alpha$ is 0~1. $\alpha = 1$ indicates a cell fully occupied by liquid; $\alpha = 0$ means that the grid cell is full of air; $\alpha = 0$~1 corresponds to an interface region with mixed phases.

The transport equation for α is:

$$\frac{\partial \alpha}{\partial t} + \nabla \cdot \left( \alpha \overline{\boldsymbol{U}} \right) + \nabla \cdot \left( \boldsymbol{u}_c \alpha (1 - \alpha) \right) = 0 \tag{14}$$

where $\overline{\boldsymbol{U}}$ is the velocity vector and $\boldsymbol{u}_c$ is the compression velocity, defined as the relative velocity between phases.

The expression for $\boldsymbol{u}_c$ is given as [37]:

$$\boldsymbol{u}_c = \boldsymbol{n}_f \min \left[ C_f \frac{|\varphi|}{|S_f|}, \max \left( \frac{|\varphi|}{|S_f|} \right) \right] \tag{15}$$

where $\varphi = \overline{\boldsymbol{U}}_f \cdot S_f$ represents the volume flux of the mesh cell; $C_f$ is the compression factor; $n_f = \frac{(\nabla \alpha)_f}{\left| (\nabla \alpha)_f + \delta_n \right|}$ denotes the face-centered estimate of the interface normal vector n.

In this study, the standard $k$–$\varepsilon$ model is combined with the VOF method for unsteady simulation of free-surface discharge flow. The SIMPLE algorithm is applied for equation discretization, while the PISO method is used for pressure–velocity coupling. Time integration is performed using a second-order implicit backward Euler method. The convection terms, volumetric flux, turbulent kinetic energy, and dissipation rate are discretized using a second-order upwind scheme, and diffusion terms are discretized using a second-order central difference scheme. The residual convergence criterion is set to $10^{-6}$.

**2.1.3 Initial and boundary conditions.** The upstream boundary adopts a uniform velocity inlet condition, comprising an upper gas inlet and a lower water inlet. The water and gas boundary conditions are automatically adjusted based on the water depth, inlet uses the boundary condition *variableHeightFlowRateInletVelocity*. The gas boundary is defined as a pressure inlet with the pressure set to zero. The downstream outlet is set as a free outflow boundary, with the outlet pressure equal to atmospheric pressure (i.e., zero gauge pressure). The bottom surface of the wide-crested weir is defined as a solid wall with a no-slip condition. The no-slip condition is also used in the spanwise direction. A vent hole of dimensions 0.03 m × 1.2 m is included in the model. Its position matches that in the experimental setup, and its boundary conditions are the same as those used for the gas phase.

Initially, there is no discharge, and the downstream flow rate is zero. As the simulation progresses, water begins to discharge from the outlet. Following the method proposed by Xue et al. [24], the consistency between the inlet and outlet flow rates is used as the criterion for determining whether a steady-state flow condition has been reached. The inlet flow rate in the simulation is calculated using the following conversion formula:

$$q = \frac{Q \times 10^{-3}}{B} \times a \tag{16}$$

where $q$ is the inflow discharge in the numerical simulation (m³/s), $Q$ is the inflow discharge used in the experimental setup (L/s), $B = 1.2$ m is the width of the experimental flume, and $a = 1.2$ m is the spanwise length of the numerical domain. The simulated inflow discharges are set to $q = 0.015$, 0.030, 0.045, 0.060, and 0.075 m³/s.

**2.1.4 Time step setting.** In OpenFOAM, the Courant number (Co) governs automatic time step adjustment. Users specify only the initial time step and a maximum Courant number; the solver then dynamically adjusts the time step throughout the simulation to satisfy the specified Co limit. This approach helps reduce computational time while maintaining numerical stability. To ensure stability, the Courant number must remain below 0.5. The Courant number is defined as [38]:

$$Co = \frac{\delta t \left| \overline{\boldsymbol{U}} \right|}{\delta x} \tag{17}$$

In the equation, $|\overline{U}|$ is the magnitude of the velocity vector; $\delta t$ is the time step, and $\delta x$ denotes the minimum grid spacing.

To ensure numerical stability during the simulation, if $|\overline{U}|$ is fixed, a smaller grid size requires a correspondingly smaller time step. To prevent numerical divergence, if the calculated Courant number exceeds the specified maximum value under the current time step setting, the time step is immediately reduced. If the computed Courant number exceeds the specified maximum at any point, the time step is immediately reduced. Conversely, when the computed Courant number remains below the limit, the time step is gradually increased to improve efficiency without compromising stability. In this study, the maximum Courant number is set to 0.4 [27]. The interFoam solver enables automatic time step control by setting adjustTimeStep to on in the controlDict file [39].

### 2.2 Model building

**2.2.1 Computational domain and grid generation.** Figs 2 and 3 illustrate the schematic diagram of the computational domain and the local grid refinement. The size of the two-phase flow domain (Fig 2) is consistent with the experimental model described in Chapter 2, and the complete numerical model is shown in Fig 3. The wide-crested weir has a width of 0.5 m, with the leading edge located at the origin of the $x$-axis. The weir height $H$ is variable, matching the experimental scheme. The model thickness in the $z$-direction is set to a = 1.2 m. To reduce computational cost, the flume length is appropriately shortened. The velocity inlet for water is positioned at $x = 0.6$ m, with flow aligned along the $x$-axis, and the pressure monitoring point is located at $x = 0.8$ m. The wide-crested weir surface has a large discharge capacity and stable flow characteristics, which reduce the occurrence of excessive negative pressure and provide economic advantages [40]. Each simulation is conducted for 60 s, and the pressure data obtained at the monitoring points demonstrate good reliability, consistency, and representativeness.

**2.2.2 Mesh independence and validation.** To evaluate the influence of mesh density on simulation accuracy and computational efficiency, mesh independence verification is performed [41]. Fig 4 presents the mesh independence analysis, where solid lines represent numerical results and dotted lines represent experimental data. Three-dimensional structured block meshes are employed, and the total mesh number is gradually increased. By refining the global mesh size and the local mesh size in the discharge region, six mesh densities are tested. The time-averaged pressure at the monitoring points is used to assess mesh sensitivity. When the refined grid size decreases from 8.35 mm × 8.35 mm to 7.15 mm × 7.15 mm, the mean pressure shows negligible variation. This demonstrates that a local grid size of 8.35 mm × 8.35 mm is sufficient for accurate simulations, as further refinement does not significantly improve accuracy but greatly increases computational cost. In the drop region, bubble diameters generated by water impingement or turbulence typically range from 1 mm to 10 mm [42]. Therefore, a grid size of approximately 8.35 mm in this region is adequate to capture bubble formation, breakup, and collapse processes. All other regions are discretized with the same mesh size.

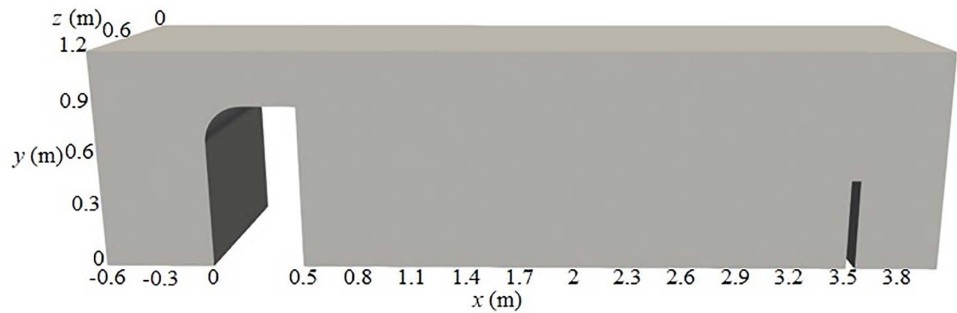

**Fig 2. Schematic diagram of computational domain.**

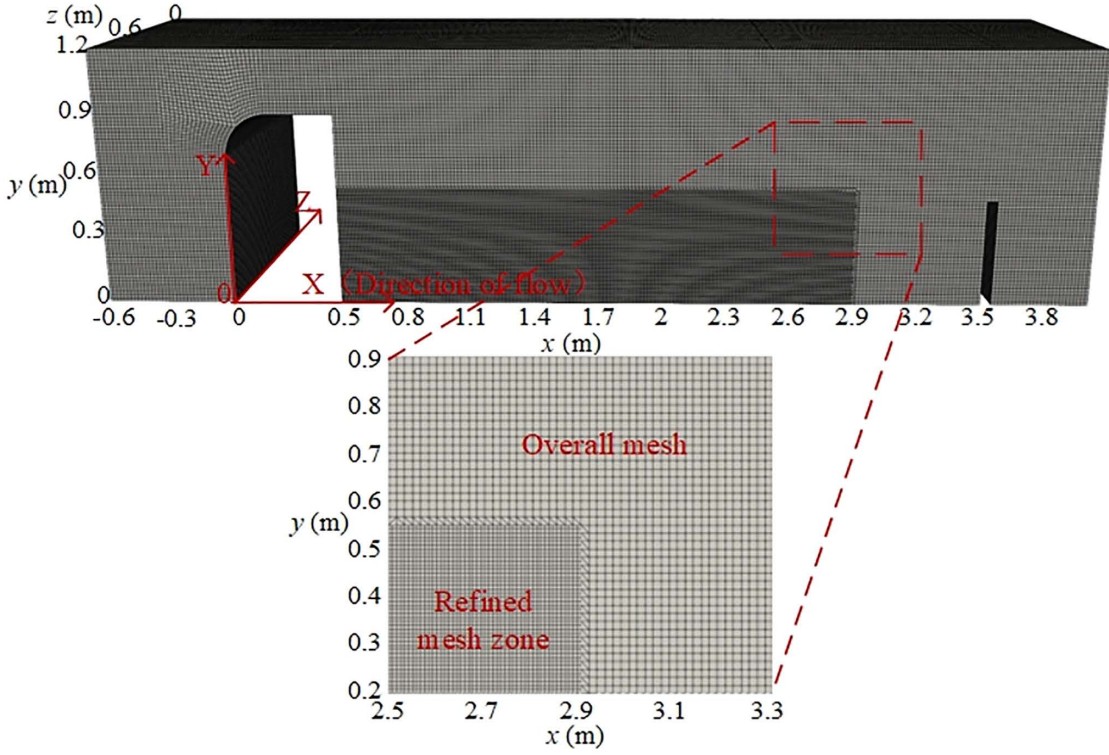

**Fig 3. Grid division scheme of computational region.**

Fig 5 compares experimental and simulated results. For a weir height $H=90$ cm and an inflow rate $Q=60$ L/s, five downstream static water depths are selected to validate the numerical method. As shown in Fig 5a, the numerical results exhibit the same trend as the experimental data, with time-averaged pressure increasing with downstream static water depth. The maximum relative error among the five cases is less than 5%, confirming that the numerical model can accurately reproduce physical behavior. Since underwater noise SPL is derived from pressure fluctuations, the numerical model is further validated by comparing simulated fluctuating pressure with experimental measurements, ensuring that the selected mesh resolution is adequate for acoustic-related analysis. As shown in Fig 5b, the shaded gray region corresponds to a ±5% error band based on the experimental measurements. The close agreement between the simulation results and the experimental data, with most simulation points falling within the 5% error band, also demonstrates the reliability and accuracy of the numerical model.

## 3 Experimental results and analysis

### 3.1 Changes in SPL

**3.1.1 Effects of flow rate, downstream water depth, and weir height on $Lpw$.** Fig 6 presents the variation in underwater noise SPL as a function of incoming flow rate and downstream static water depth, with the wide-crested weir height held constant. The SPL ranges from 105 dB to 180 dB. At a fixed downstream static water depth and weir height, the SPL increases with the flow rate, reaching its maximum at $Q=75$ L/s. This trend is consistent across different flow conditions. Furthermore, for a given flow rate and weir height, the SPL decreases with increasing downstream static water depth, with the maximum values consistently occurring at $h=10$ cm. Comparison of Figs 6a through 6e reveals that, under fixed flow rate and downstream static water depth, the SPL increases with weir height, with peak values observed at $H=90$ cm.

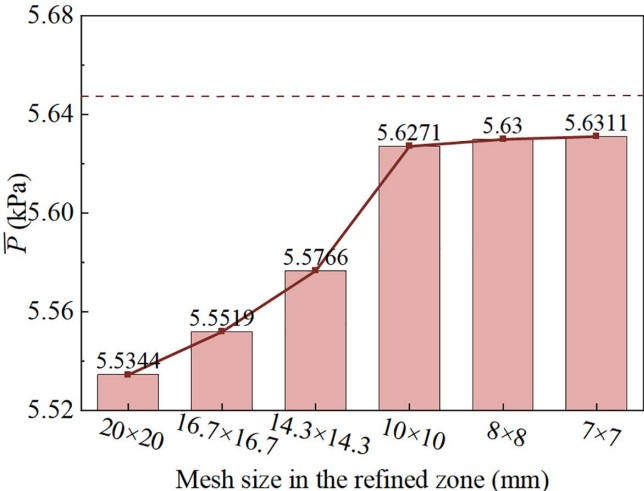

**Fig 4. Grid independence analysis.**

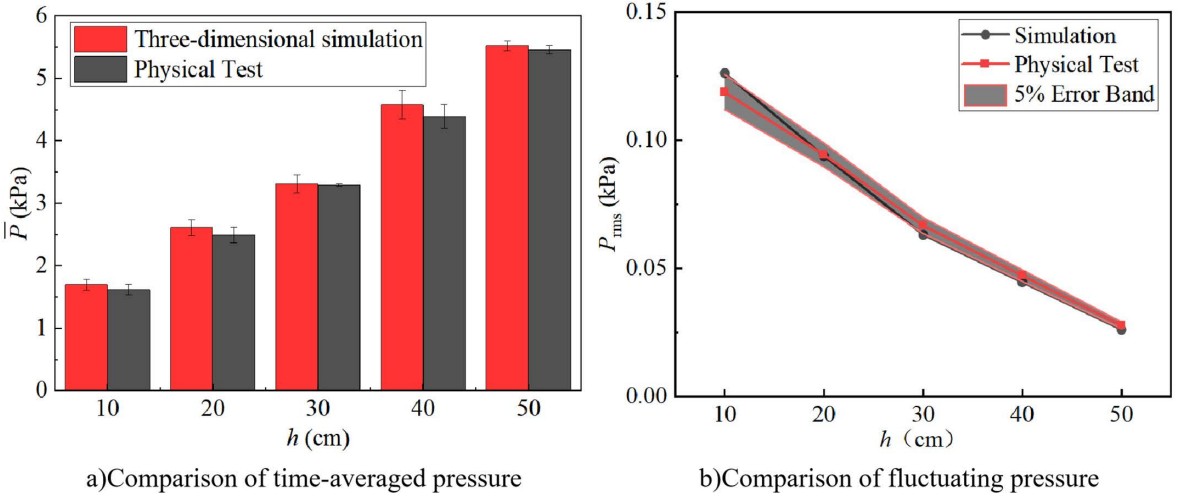

a)Comparison of time-averaged pressure

b)Comparison of fluctuating pressure

**Fig 5. Comparison of experimental and calculated results.**

### 3.1.2 Effects of flow rate, downstream water depth, and weir height on fluctuating pressure SPL. Fig 7

illustrates the relationship between fluctuating pressure SPL and both inflow rate and downstream static water depth, with
weir height fixed. The fluctuating pressure level ranges from 105 dB to 180 dB, consistent with underwater noise levels.
Both increase with inflow rate, and both decrease with increasing downstream static water depth, with the maximum
consistently observed at $Q = 75$ L/s and $h = 10$ cm. Comparison of Figs 7a–7e further shows that, at fixed inflow and water
depth, the fluctuating pressure level increases with weir height, with the maximum observed at $H = 90$ cm.

## 3.2 Correlation analysis of underwater noise

The Pearson correlation coefficient ($r$) was applied to analyze relationships among underwater noise SPL, fluctuating
pressure SPL, wide-crested weir height, incoming flow rate, and downstream static water depth. Both underwater noise

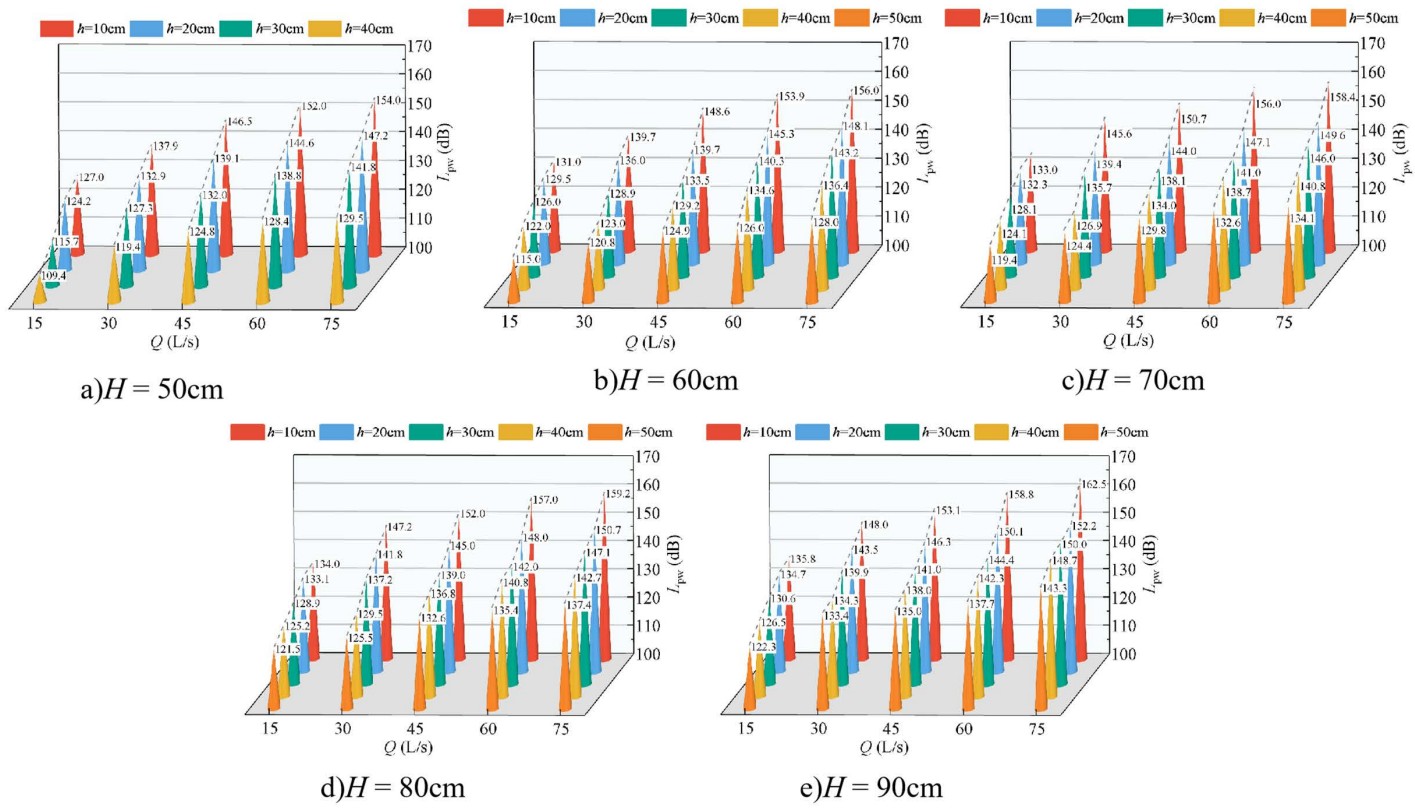

**Fig 6. Relationship between flow rate, water depth, and underwater noise SPL at various weir heights.**

SPL and fluctuating pressure SPL are expressed in dimensionless form and require no further normalization. Pearson correlation analysis is used to quantify the strength of association between variables, while linear regression between $L_{pw}$ and fluctuating pressure SPL is applied to evaluate their quantitative relationship and provide a basis for mechanistic interpretation.

Table 1 presents the correlation results. Both underwater noise and fluctuating pressure SPLs show highly significant positive correlations with wide-crested weir height and incoming flow rate, and significant negative correlations with downstream static water depth. For underwater noise SPL, the order of correlation strength is: downstream static water depth ($|r| = 0.666$)> incoming flow rate ($|r| = 0.647$)> wide-crested weir height ($|r| = 0.290$). For fluctuating pressure SPL, the order is: downstream static water depth ($|r| = 0.701$)> incoming flow rate ($|r| = 0.575$)> wide-crested weir height ($|r| = 0.246$).

Fig 8 shows the correlation between underwater noise SPL and fluctuating pressure SPL. The Pearson correlation coefficient is $r = 0.88$, indicating a strong positive correlation. The corresponding $p$-value is less than 0.05, suggesting that fluctuating pressure significantly affects underwater noise levels. This result indicates that fluctuating pressure SPL can serve as an indicator for underwater noise variation. The fitted relationship is: $L_{pw} = 1.23\text{SPL} - 61.4$.

### 3.3 Analysis of frequency characteristics of underwater noise

**3.3.1 Effect of downstream static water depth on frequency.** Fig 9 shows the time–frequency spectrum and Fourier transform spectrum of underwater noise at different downstream static water depths, with weir height $H = 80\,\text{cm}$ and inflow rate $Q = 30\,\text{L/s}$. To improve readability, only representative FFT spectra are shown in the main text, while full-resolution

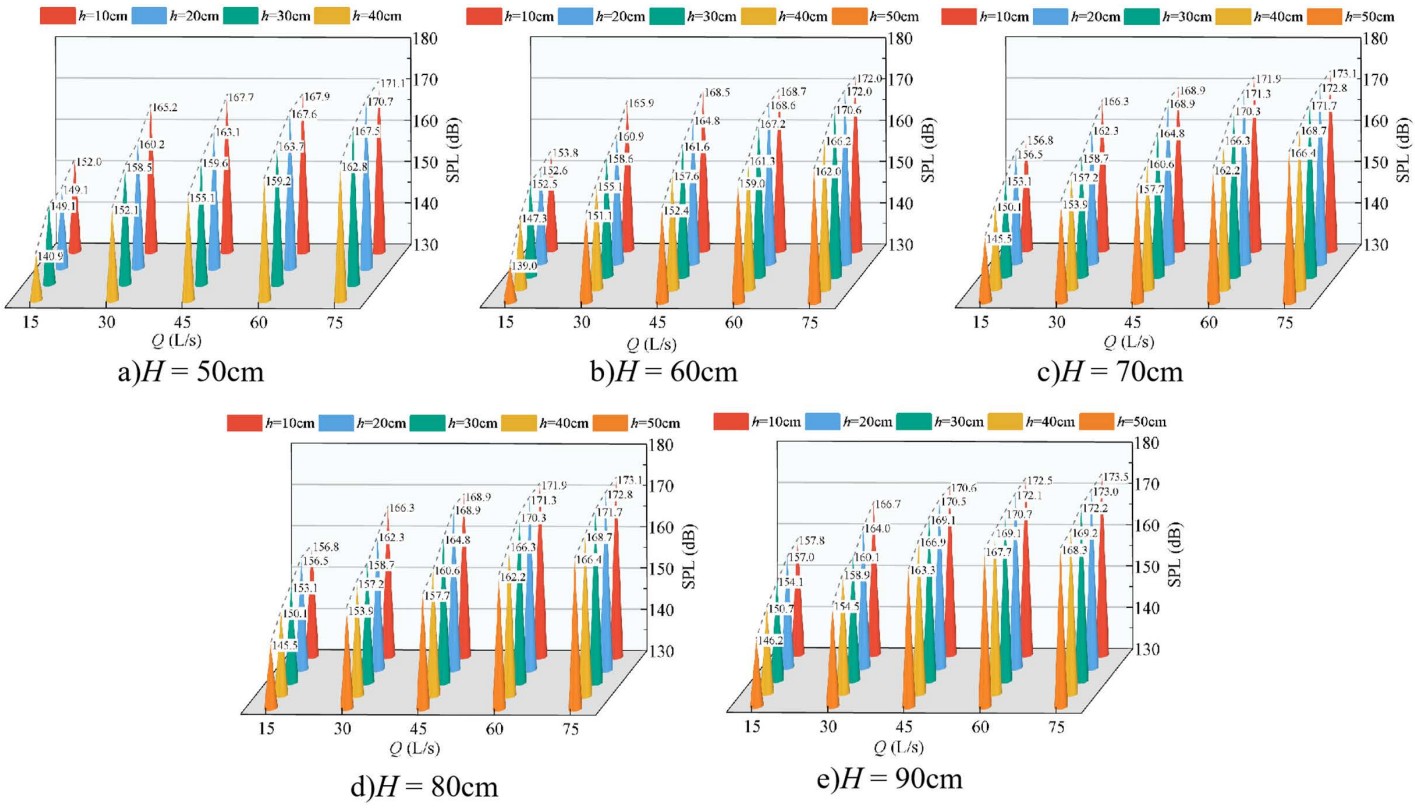

**Fig 7. Relationship between flow rate, water depth, and fluctuating pressure SPL at various weir heights.**

spectra for all experimental conditions are provided in the Supporting Information. Across the five depth conditions, the main frequencies are concentrated in the ranges 0–20 Hz and 150~400 Hz. The noise is primarily composed of low-frequency components generated by turbulence and medium-frequency components generated by discharge flow. Under $H=80$ cm and $Q=30$ L/s, the dominant frequencies appear in the low- and mid-frequency ranges, with strong peaks near 0~20 Hz, 200 Hz, 350 Hz, and 500 Hz. When the downstream static water depth is shallow ($h=40$ cm~50 cm,), the dominant noise remains at low frequency, with energy concentrated at 0–20 Hz. At shallow depths, water discharged from the weir strongly impacts the flume bottom and generates strong vortices. As depth increases, this impact weakens [43–44]. This indicates that turbulence is the dominant source of noise, while bubbles and shock contribute as secondary sources.

Taking $H=80$ cm, $Q=30$ L/s, and $h=30$ cm as an example, wavelet transform analysis shows that in the time range of 0.1–0.6 s, energy is significantly higher at 0–20 Hz, 200 Hz, and 350 Hz, while it is relatively weaker at other times. The 0–20 Hz component represents the first dominant period, with low frequency and long-period fluctuations. The ~200 Hz component represents the second dominant period, with medium frequency and short-period fluctuations. In the time–frequency spectrum, the 200 Hz component undergoes four distinct cycles of alternating intensity within 1 s, reflecting stable periodic oscillation behavior. A weaker peak also appears near 500 Hz in the 0.4~1.0 s time window, but this high-frequency component attenuates more rapidly. In general, higher frequencies exhibit faster energy dissipation, showing that high-frequency signals decay in a shorter time. These time–frequency characteristics are consistent with the Fourier spectra, confirming and supplementing the wavelet analysis.

**Table 1. Correlation coefficients between underwater noise, fluctuating pressure SPL, and influencing factors.**

|  | $L_{pw}$ /dB | SPL/dB | H/cm | Q/(L/s) | h/cm |
|---|---|---|---|---|---|
| **$L_{pw}$ /dB** | 1 |  |  |  |  |
| **SPL/dB** | 0.881** | 1 |  |  |  |
| **H/cm** | 0.290** | 0.246** | 1 |  |  |
| **Q/(L/s)** | 0.647** | 0.575** | / | 1 |  |
| **h/cm** | −0.666** | −0.701** | / | / | 1 |

Note: ** indicates significant correlation at 0.01 level.

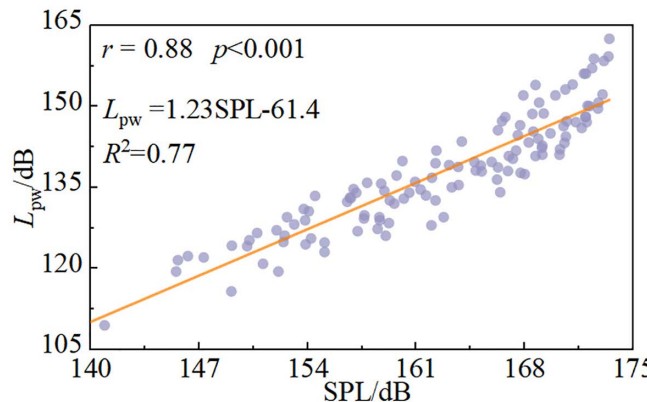

**Fig 8. Correlation between underwater noise SPL and fluctuating pressure SPL.**

**3.3.2 Influence of incoming flow rate on frequency.** Fig 10 presents the time–frequency spectrum and Fourier transform spectrum of underwater noise at different inflow rates, with fixed weir height $H = 80$ cm and downstream static water depth $h = 30$ cm. Across the five flow conditions ($Q = 15$–$75$ L/s), the dominant frequencies remain concentrated in the middle and low frequency bands, mainly at 0–20 Hz, 200 Hz, and 350 Hz. At small inflow rates, the discharge exerts weaker impact on the flume bottom, resulting in lower dominant frequency energy, with low-frequency turbulence prevailing. As inflow rate increases, stronger air entrainment occurs, leading to bubble formation, growth, and collapse, which generate additional high-frequency noise.

Taking $H = 80$ cm, $Q = 45$ L/s, and $h = 30$ cm as an example, wavelet transform analysis shows that within 1 s, energy in the ranges 0～50 Hz and ～300 Hz is elevated, displaying strong fluctuation characteristics. The 300 Hz component exhibits about 30 alternating cycles of strong and weak intensity, demonstrating more evident periodic behavior, whereas other frequency bands show lower energy. This phenomenon corresponds to transient turbulent structures and localized bubble collapse, which rapidly release and dissipate high-frequency energy. Overall, as inflow rate increases, turbulence and impact become the dominant noise sources, while bubble dynamics play a secondary role.

**3.3.3 Effect of the height of wide-crested weir on frequency.** Fig 11 presents the time–frequency spectrum and Fourier transform spectrum of underwater noise at different weir heights, with inflow rate $Q = 30$ L/s and downstream static water depth $h = 30$ cm＝. Across the five weir heights (50–90 cm), the dominant frequencies remain concentrated in the low- and mid-frequency ranges, with maxima appearing at 0–20 Hz and 200–400 Hz. When the weir height is relatively low, the discharge exerts a weaker impact on the flume bottom, and the dominant frequency energy in the time–frequency

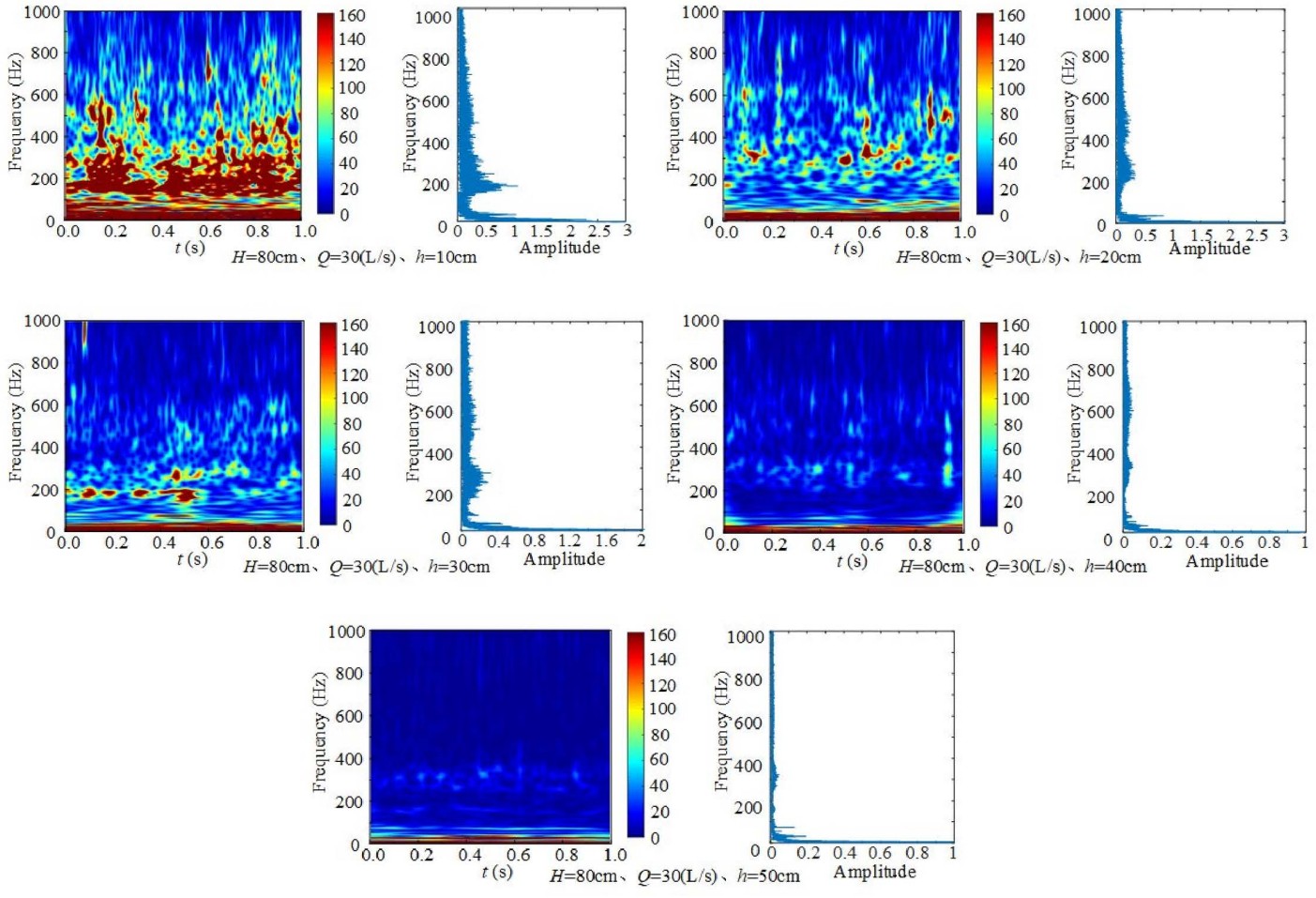

**Fig 9. Time–frequency spectra of underwater noise at different downstream static water depths (*H*=80 cm, *Q*=30 L/s).**

spectrum is correspondingly weaker. As the weir height increases, more air is entrained into the water, generating bubbles. The formation, development, and collapse of these bubbles contribute to additional high-frequency noise, making the higher-frequency components more pronounced. Within 1 s, frequency components near 0~20 Hz, 212~376 Hz, and 600 Hz exhibit higher energy and stronger fluctuations, while other periods show relatively lower energy. The 0~20 Hz band is identified as the first dominant frequency, characterized by low-frequency, long-period oscillations. The 212~376 Hz– range represents the second dominant frequency, with medium-frequency, shorter-period oscillations. At ~250 Hz, underwater noise shows strong temporal variation. In addition, frequencies in the 400~600 Hz range exhibit strong but short-lived energy, with rapid attenuation. This reflects the inherent dissipation of high-frequency signals over short timescales.

With increasing inflow rate and weir height, the energy at corresponding frequencies strengthens, and the contribution of high-frequency bands gradually rises. In combination with the findings in Section 4.3, it is evident that low-frequency noise is primarily governed by large-scale turbulent vortices. These vortices play a critical role in turbulent energy transfer, momentum transport, water entrainment, and mixing. Their occurrence is often accompanied by flow separation, during which velocity gradients intensify and fluid motion becomes irregular. This instability results in pronounced fluctuations in the frequency of underwater noise [45].

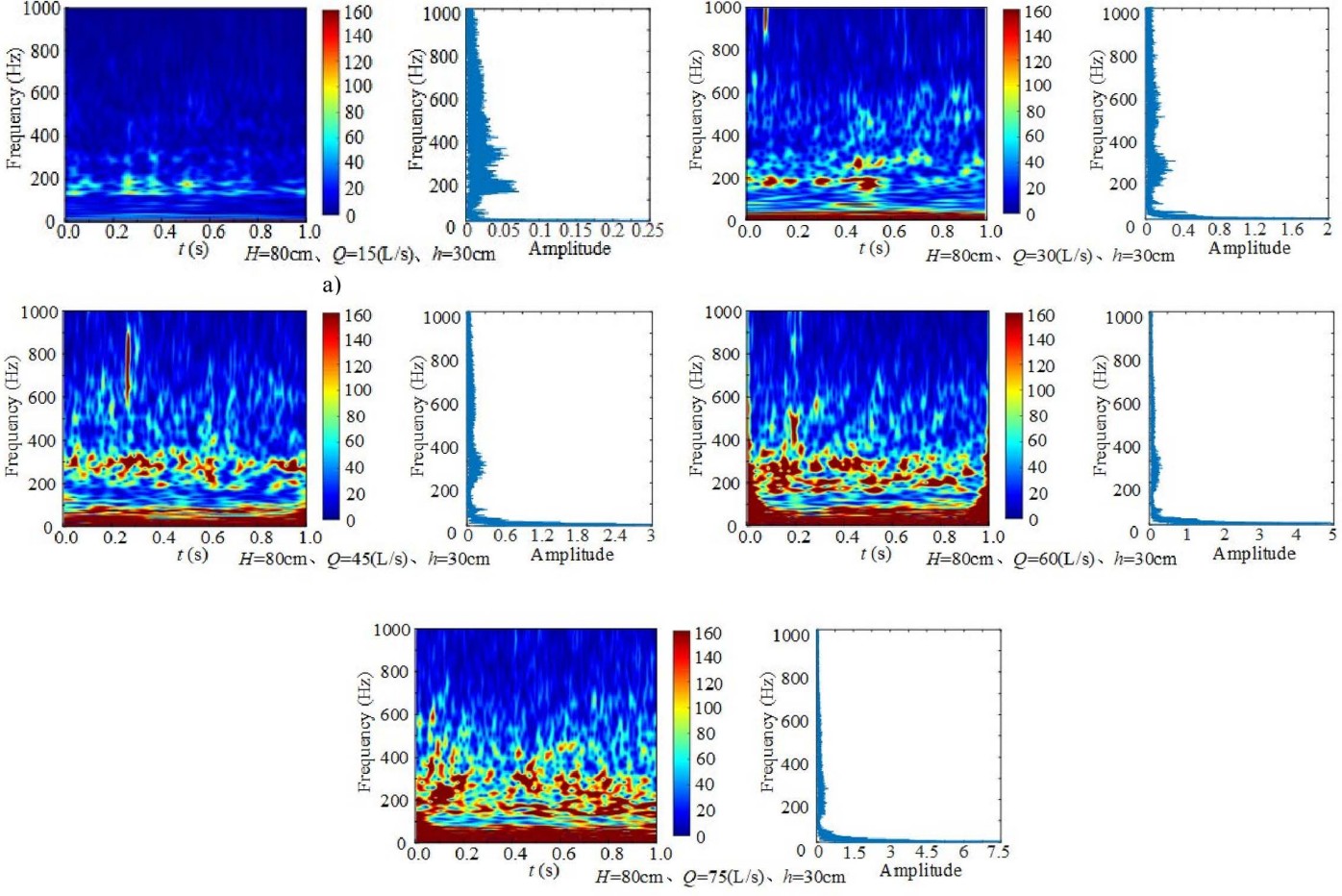

**Fig 10. Time–frequency spectra of underwater noise at different inflow rates ($H = 80$ cm, $h = 30$ cm).**

## 4 Analysis of underwater noise mechanism

The characteristics of bubble evolution, turbulent kinetic energy, vortex distribution, flow expansion, mean-square distribution of fluctuating pressure, and time–frequency domain behavior are examined in this chapter to reveal the mechanism of total SPL and main frequency variation presented in Sections 3.1 and 3.3.

### 4.1 Fluid regime change

Fig 12 shows the instantaneous flow state of the bubble–water mixed flow with a free surface under $H = 90$ cm, $Q = 75$ L/s, and $h = 50$ cm. As shown in Fig 12, when the main water tongue is injected into the downstream water body, its velocity exceeds that of the surrounding fluid, forming a shear zone that promotes instability and vortex generation. Simultaneously, disturbance and tumbling occur at the water–air interface, creating a bubble–water mixed flow with a free surface. The impact force generated by the main water tongue is transmitted through the water to the tank bottom. Shape changes and breakup of vortex structures induce pressure fluctuations, which propagate through the fluid and act on the pressure measurement point [44]. Bubbles rise gradually under buoyancy until rupturing at the water–air interface, releasing instantaneous high-intensity radiated sound waves that affect the fluctuating pressure

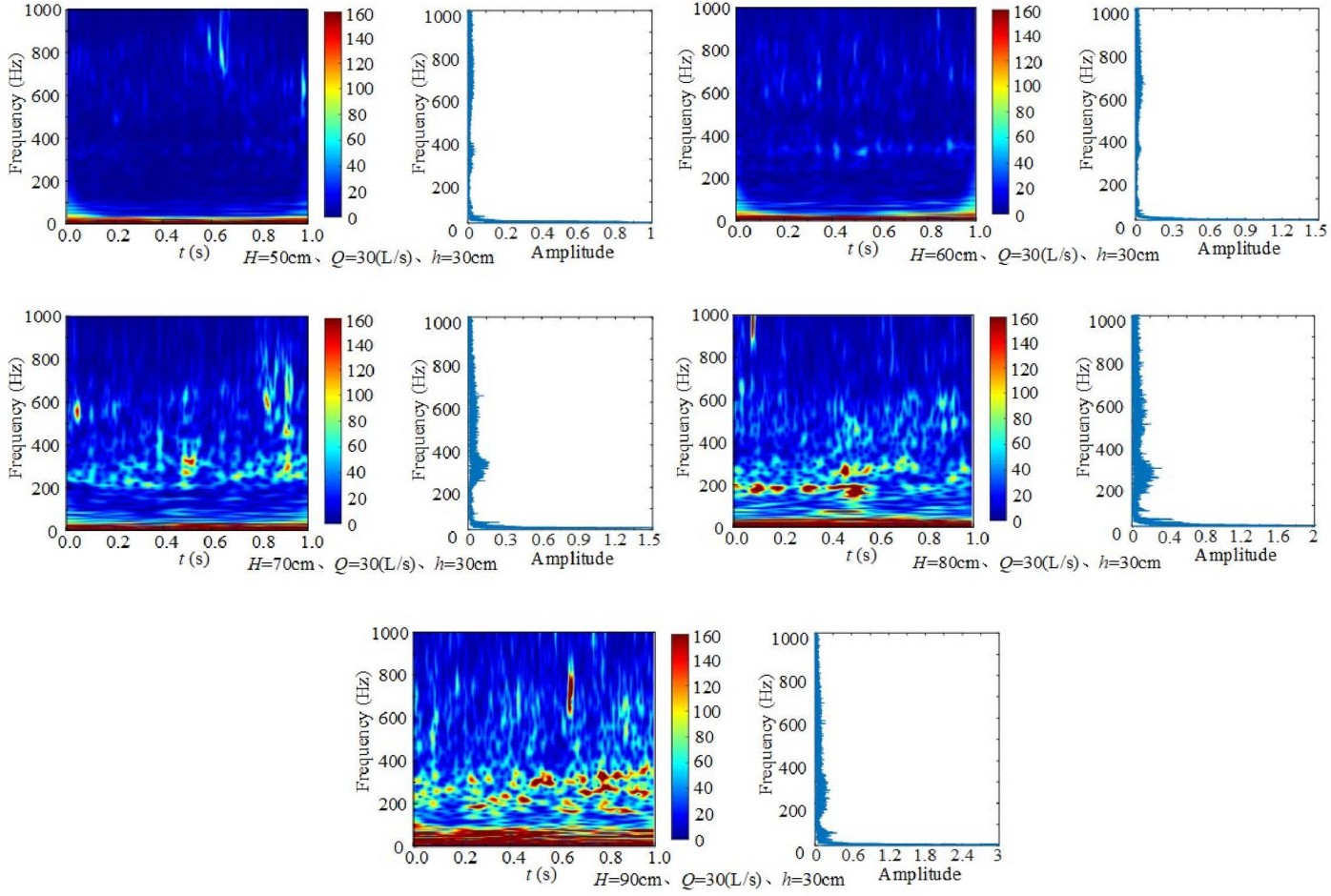

**Fig 11. Time–frequency spectra of underwater noise at different weir heights ($Q = 30$ L/s, $h = 30$ cm).**

at the measurement point. Therefore, three processes contribute to fluctuating pressure: the impact of the discharged water flow on the downstream water body, the morphological change and breakup of vortex structures, and bubble rise followed by rupture at the water–air interface [2].

To further examine bubble evolution in the bubble–water mixed flow, Fig 13 presents the flow pattern during $t = 4.9s \sim 6.0$ s under the same conditions. From $t = 4.9s \sim 5.3s$ and $t = 5.7s \sim 6.0$ s, entrained gas becomes isolated from the atmosphere and larger bubbles are formed, which are stretched by shear forces and split into several smaller bubbles. Between $t = 5.4$ s $\sim 5.7$ s, small bubbles rise under buoyancy, their shape and volume varying with pressure changes. During this process, multiple small bubbles approach and coalesce into larger bubbles. Between $t = 5.0$ s $\sim 5.5$ s, bubbles reach the water–air interface, burst, and release energy, resulting in a reduction in bubble volume and local pressure variation. At the moment of bursting, bubbles emit noise [30]. This released energy propagates through the water as radiated sound waves, influencing the fluctuating pressure at the measurement point [16]. Combining Figs 5−1 and 5 -2, it is evident that the mainstream water tongue entering the downstream water body induces vortex structural changes and generates a bubble–water mixed flow. Fluctuating pressure is therefore affected by the discharge flow impact, pressure changes associated with vortex breakup, and sound waves generated by bubble collapse.

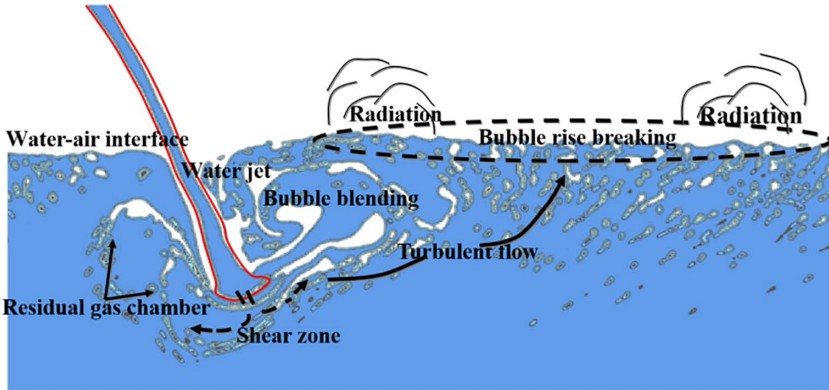

**Fig 12. Instantaneous flow pattern of bubble–water mixed flow with free surface under *H*=90 cm, *Q*=75 L/s, *h*=50 cm.**

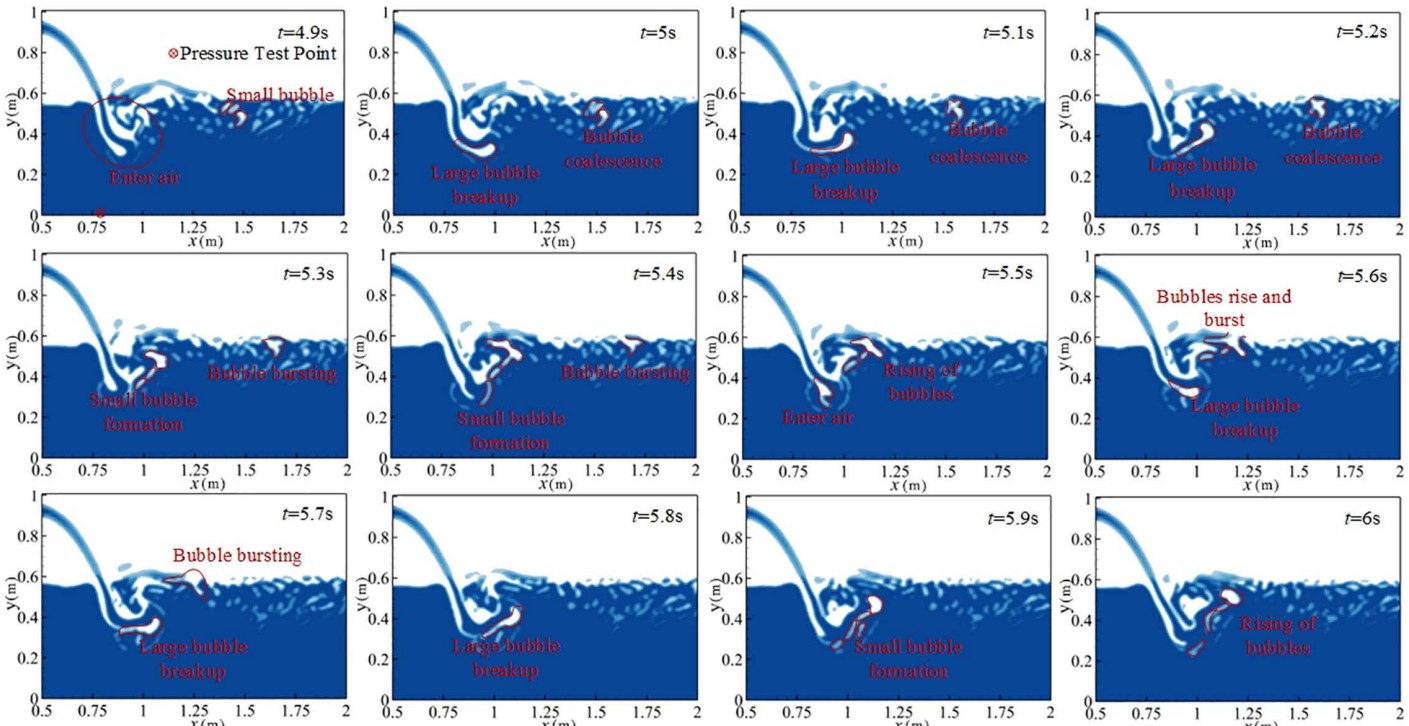

**Fig 13. Flow pattern changes at *H*=90 cm, *Q*=75 L/s, *h*=50 cm during *t*=12.7–13.8 s.**

## 4.2 Effect of turbulent kinetic energy on underwater noise

Turbulent kinetic energy (*k*) is a parameter that characterizes turbulence intensity, directly reflecting the flow turbulence state and playing a central role in energy conversion and transfer [46]. It is defined as:

$$k = \frac{1}{2}\left((u'u') + (v'v') + (w'w')\right)$$

(18)

In the equation, $u'$ is fluctuating flow velocity in the $x$-direction; $v'$ is fluctuating flow velocity in the $y$-direction; $w'$ is fluctuating flow velocity in the $z$-direction.

**4.2.1 Effect of downstream static water depth on turbulent kinetic energy.** Fig 14 shows the effect of downstream static water depth on the vertical distribution of $k$ under $Q=75$ L/s and $H=90$ cm when the $x$-axis position is 0.8 m. When the downstream static water depth is 10 cm~20 cm, turbulent kinetic energy in the discharge flow impact area is high, particularly at the flume bottom. At this depth, mixing is pronounced, and disturbances at the water–air interface and underwater impact point are more intense, exerting a strong influence on fluctuating pressure at the measuring point. When the downstream static water depth increases to 30~50 cm, the maximum turbulent kinetic energy shifts upward from the bottom toward the surface, the distribution becomes more uniform, and the overall energy level decreases, although local high-value zones persist. Turbulent kinetic energy at the bottom decreases progressively, leading to a reduction in underwater noise.

Fig 15 presents the radial distribution of turbulent kinetic energy at different downstream static water depths under $Q=75$ L/s and $H=90$ cm when the $z$-axis position is 0.6 m. Turbulent kinetic energy is mainly distributed in the shear layer between the main jet and the surrounding fluid, namely the impact zone behind the wide-crested weir. When the downstream static water depth is 10 cm~20 cm, the influence range of the main jet extends between $x=0.7$ m and $x=1.6$ m. Strong interactions between the discharged water and downstream fluid increase turbulent kinetic energy and pressure fluctuations at the measuring point. The vortices generated by mixing continuously transfer turbulent kinetic energy downstream until it is gradually dissipated. At 30~50 cm depths, both the range and intensity of turbulent kinetic energy are reduced, becoming concentrated near the surface, while the effect on the tank bottom is weakened, resulting in lower underwater noise.

Fig 16 shows the spanwise distribution of turbulent kinetic energy at 0.1 m below the water surface along the $y$-axis for different downstream static water depths under $Q=75$ L/s and $H=90$ cm. At a fixed $x$-coordinate, variation of turbulent kinetic energy along the $z$-axis is not pronounced. Vortices formed during mixing continue transferring turbulent kinetic energy downstream until dissipation occurs. At 10~20 cm depths, turbulent kinetic energy on the section changes only slightly with increasing inflow. Its distribution evolves from locally concentrated to more uniform, while overall intensity decreases and dissipation accelerates, leading to reduced underwater noise.

**4.2.2 Effect of incoming flow rate on turbulent kinetic energy.** Fig 17 shows the effect of incoming flow rate on the vertical distribution of $k$ under $h=20$ cm, $H=80$ cm, and $x$-axis position of 0.8 m. In the vertical direction, the influence of the discharged flow is most pronounced at the water surface, where turbulent kinetic energy is relatively high and exhibits clear stratification. This is because the surface layer is more susceptible to shear forces and mixing effects, whereas the

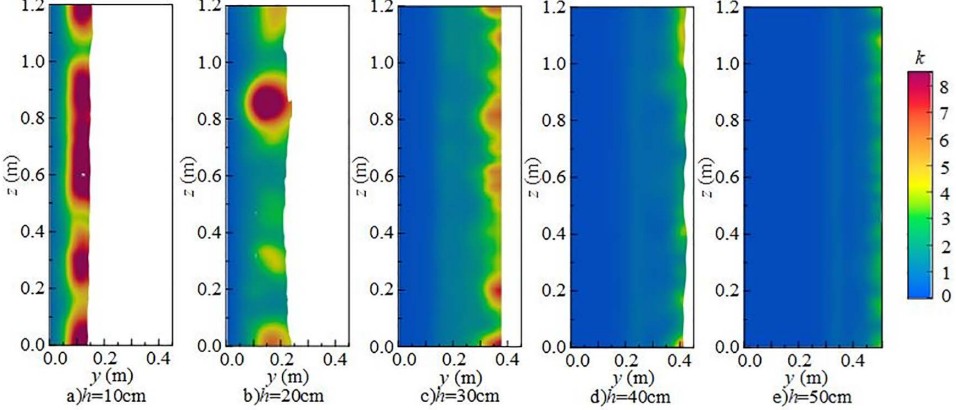

**Fig 14. Vertical variation of turbulent kinetic energy with different downstream static water depths under $Q=75$L/s and $H=90$ cm.**

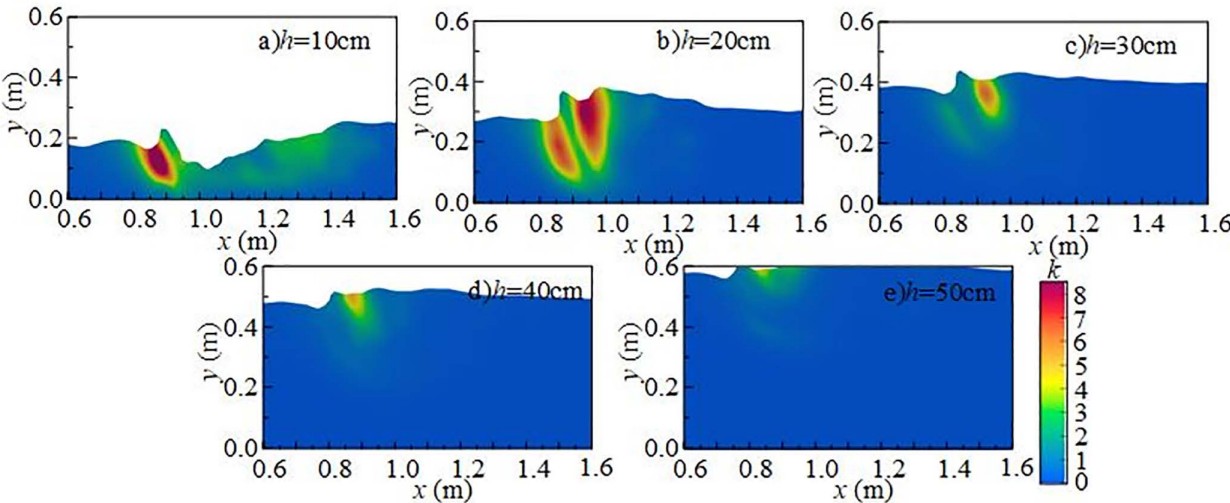

**Fig 15. Radial variation of turbulent kinetic energy with different downstream static water depths under *Q*=75 L/s and *H*=90 cm.**

bottom layer is less affected by turbulence from the upper flow and is buffered by the surrounding water body, resulting in a more gradual variation of turbulent kinetic energy. When the flow rate increases from 45 to 75 L/s, mixing intensifies, and both the range and intensity of turbulent kinetic energy increase markedly. Stronger disturbances extend into the downstream region, significantly affecting pressure at the measuring point. Turbulent kinetic energy at the flume bottom also rises with increasing inflow, leading to stronger pressure fluctuations.

Fig 18 presents the radial distribution of turbulent kinetic energy under *h*=20 cm, *H*=80 cm, and *z*-axis position of 0.6 m. Turbulent kinetic energy is primarily concentrated in the shear layer between the main jet and the surrounding water, with maximum values in the impact region behind the wide-crested weir. The distribution is clearly non-uniform. At flow rates of 15~30 L/s, the jet impact is weak, turbulent kinetic energy is low, and the influence range is limited. Energy transfer is restricted and dissipates rapidly. At higher flow rates of 45~75 L/s, jet impact and mixing are intensified, significantly increasing turbulent kinetic energy and enlarging its influence range along the *x*-axis. With greater disturbance and energy transfer distance, pressure fluctuations at the measuring point increase, thereby amplifying underwater noise.

Fig 19 shows the spanwise distribution of turbulent kinetic energy at 0.1 m below the water surface along the *y*-axis under *h*=20 cm and *H*=80 cm. At fixed *x*-coordinates, variation of turbulent kinetic energy along the *z*-axis is relatively small, indicating stable energy transfer and a nearly uniform distribution in this direction. At flow rates of 15~30 L/s, turbulent kinetic energy changes only slightly across the section, with weak flow disturbances. At higher flow rates, downstream disturbances become stronger, turbulence effects are enhanced, and underwater noise generation and propagation intensify accordingly.

**4.2.3 Effect of wide-crested weir height on turbulent kinetic energy.** Fig 20 shows the effect of wide-crested weir height on the vertical distribution of *k* under *Q*=75 L/s, *h*=10 cm, and *x*=0.8 m. The vertical distribution along the *y*-axis is uneven. At weir heights of 50~60 cm, turbulent kinetic energy is scattered and relatively weak. At higher weir heights of 70~90 cm, mixing intensifies, and overall turbulent kinetic energy increases, particularly near the water surface, where it has a strong influence on underwater noise.

Fig 21 presents the radial distribution of turbulent kinetic energy under *Q*=75 L/s, *h*=10 cm, and *z*=0.6 m. Turbulent structures are concentrated in the shear layer between the main jet and the surrounding water, with maximum values occurring in the impact zone downstream of the weir. At a weir height of 50 cm (high discharge, low water depth), the jet–downstream interaction is weak, and turbulent kinetic energy is low. The influence range extends from *x*=0.7 m to 1.6 m,

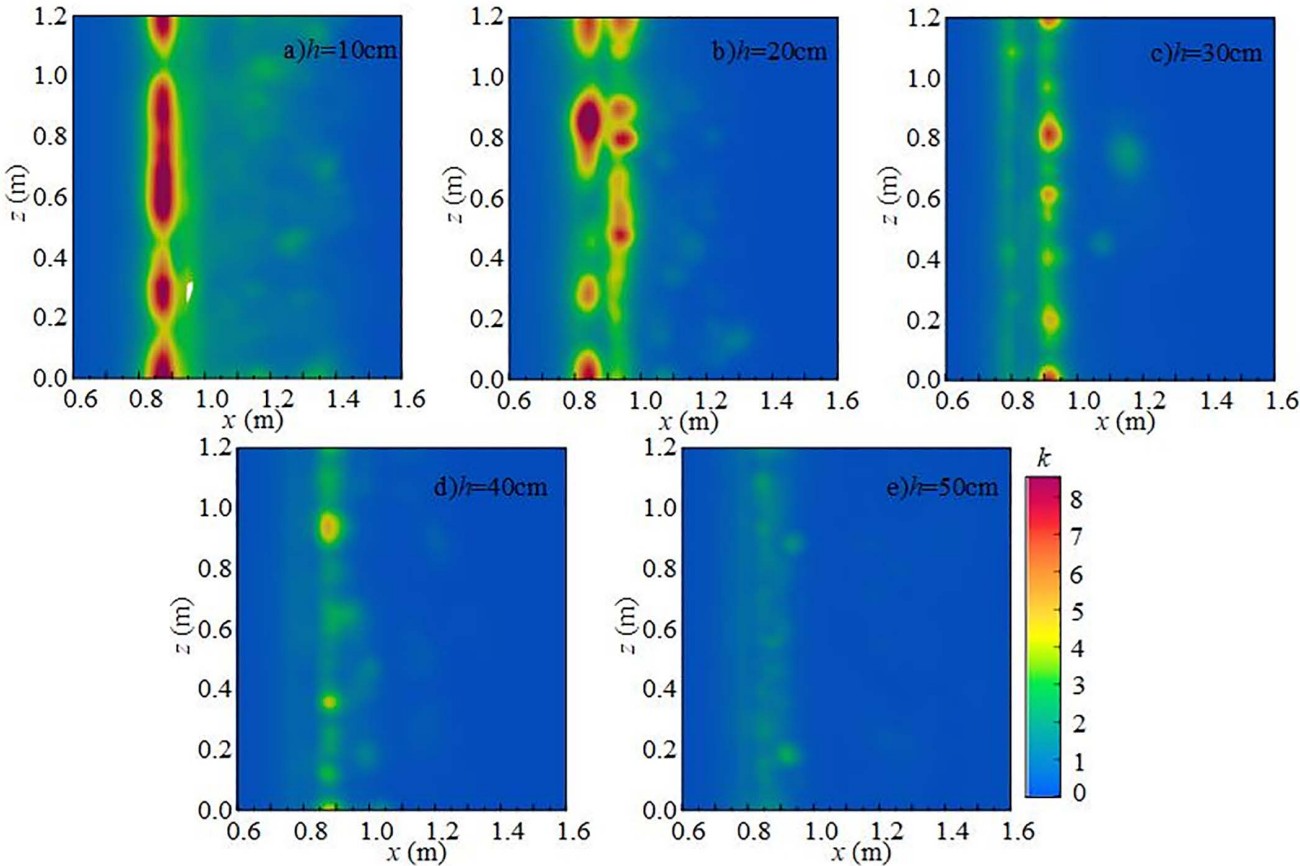

**Fig 16. Spanwise variation of turbulent kinetic energy with different downstream static water depths under *Q*=75 L/s and *H*=90 cm.**

with energy concentrated in 0.8 m~1.0 m. As weir height increases, turbulent kinetic energy intensifies, high-value regions expand, and the influence on the flume bottom becomes stronger.

At fixed *x*-values, turbulent kinetic energy changes little with the *z*-axis, indicating relatively uniform distribution in this direction. Vortices formed during mixing drive turbulent kinetic energy downstream until dissipation occurs. As weir height increases, the overall intensity of turbulent kinetic energy rises, demonstrating that higher weirs generate stronger disturbances in the downstream water body, thereby enhancing turbulence and increasing underwater noise Fig 22.

In summary, under *Q*=75 L/s and *h*=10 cm, the main influence range of turbulent kinetic energy on the flume bottom extends from 0.7 m to 1.6 m, concentrated in 0.8 m~1.0 m. As wide-crested weir height increases, turbulent kinetic energy intensifies, and the influence range expands. Bubble mixing becomes more prominent, and bubble rising and rupture further promote turbulence development. Vortices transfer turbulent kinetic energy downstream until it dissipates. According to Ref. [24], low-frequency noise originates primarily from turbulence, medium-frequency noise from water flow impacting the flume bottom, and high-frequency noise from bubble rupture. With increasing downstream static water depth, turbulent kinetic energy decreases, high-value regions shrink, and influence on the flume bottom weakens. Mixing gradually diminishes, and the contribution of bubble generation, evolution, and breakup to noise decreases. The maximum turbulent kinetic energy region shifts from the bottom to the top of the water body, and distribution becomes more uniform. Overall, total underwater SPL decreases, and the energy of low-, medium-, and high-frequency noise is

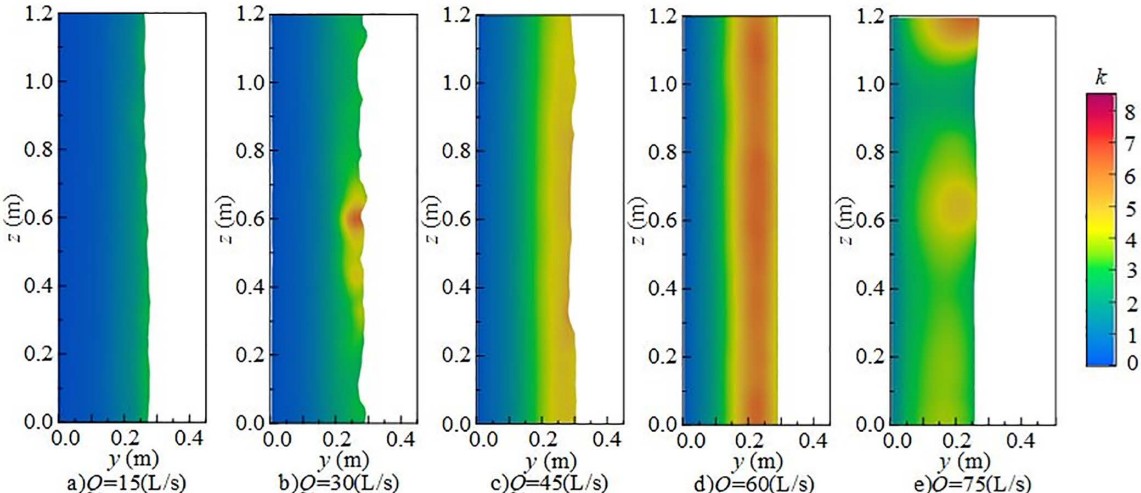

**Fig 17. Vertical variation of turbulent kinetic energy with different incoming flow rates under $h = 20$ cm and $H = 80$ cm.**

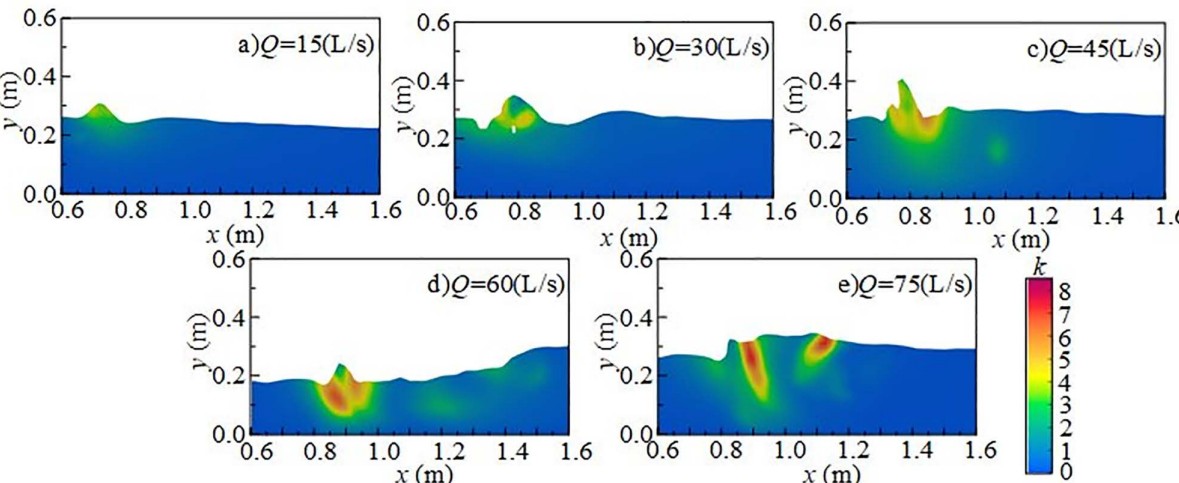

**Fig 18. Radial variation of turbulent kinetic energy with different incoming flow rates under $h = 20$ cm and $H = 80$ cm conditions.**

reduced. Low-frequency noise remains the dominant source, while medium- and high-frequency contributions decline progressively.

## 4.3 Characteristics of vortex structure in flow field

To visualize the transient flow field during discharge over a wide-crested weir, the $Q^*$ criterion is adopted to identify the spatial vortex structures. The criterion is defined as [40]:

$$Q^* = \frac{1}{2}\left(\overline{\Omega}_{ij}\overline{\Omega}_{ij} - \overline{S}_{ij}\overline{S}_{ij}\right)$$

$$(19)$$

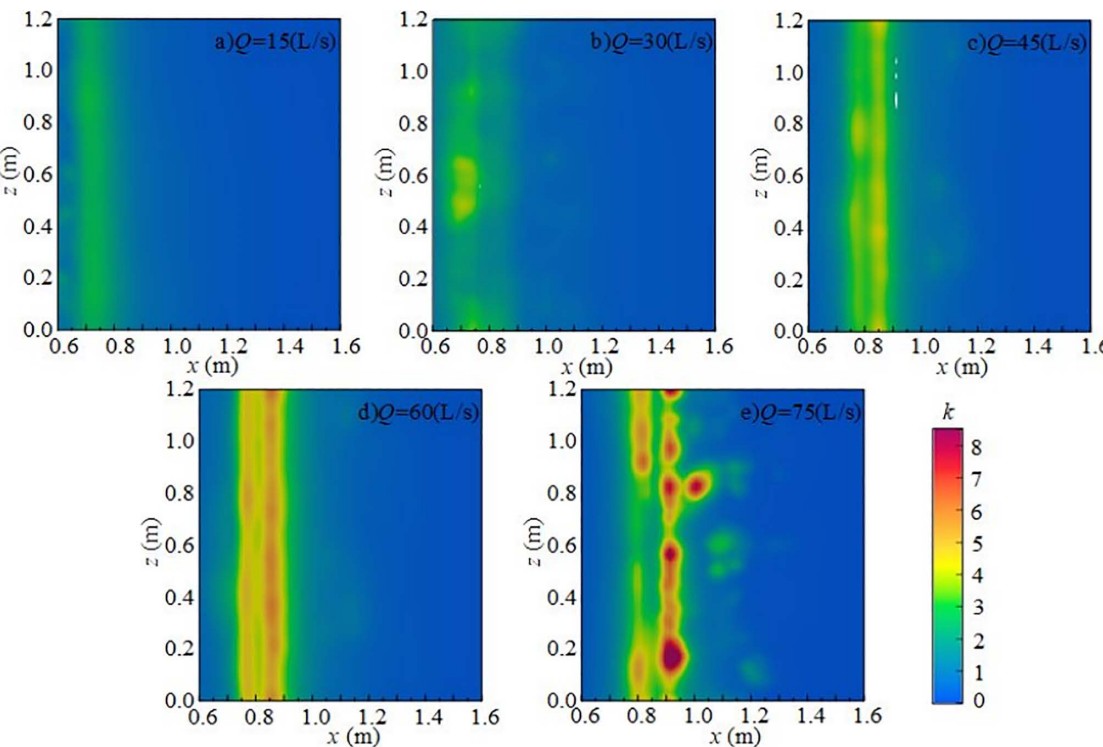

**Fig 19. Spanwise variation of turbulent kinetic energy with different incoming flow rates under $h = 20\,\text{cm}$ and $H = 80\,\text{cm}$.**

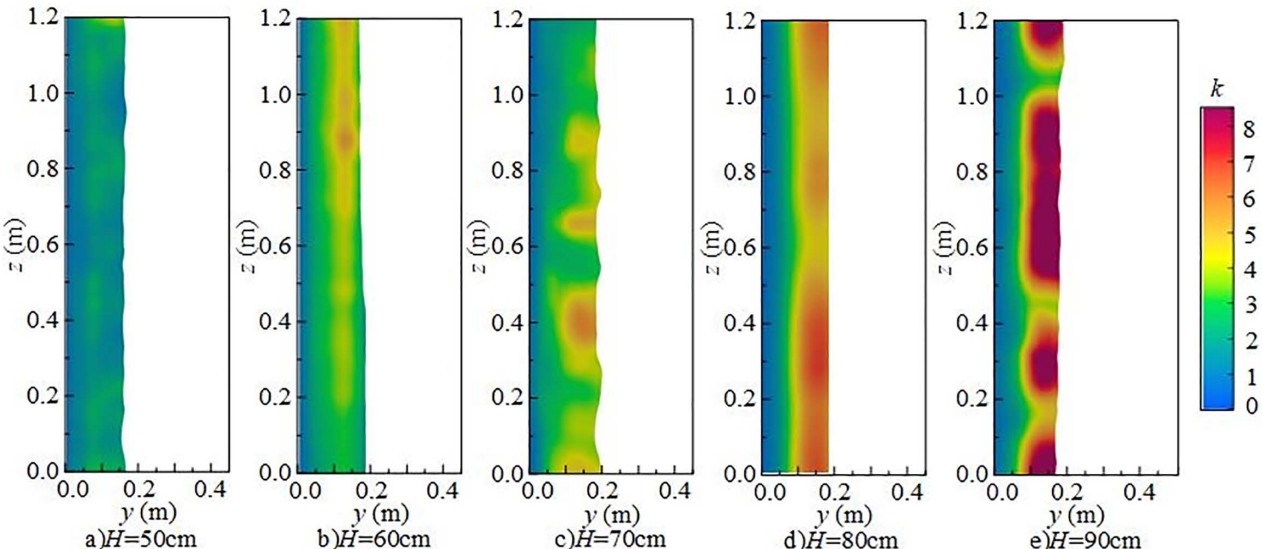

**Fig 20. Vertical variation of turbulent kinetic energy with different wide-crested weir heights under $Q = 75\,\text{L/s}$ and $h = 10\,\text{cm}$.**

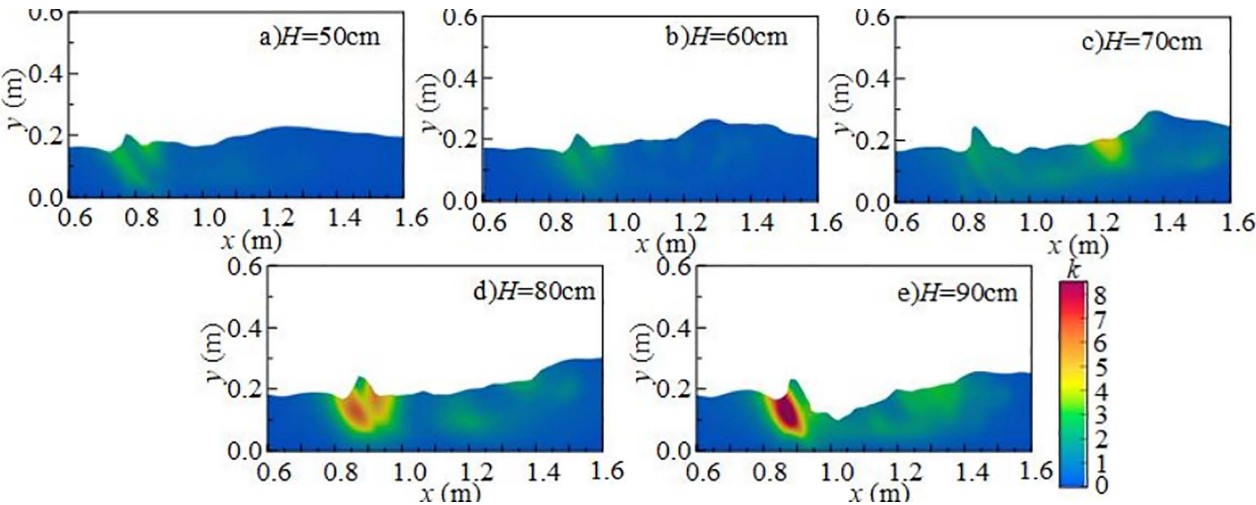

**Fig 21. Radial variation of turbulent kinetic energy with different wide-crested weir heights under $Q=75$ L/s and $h=10$ cm.**

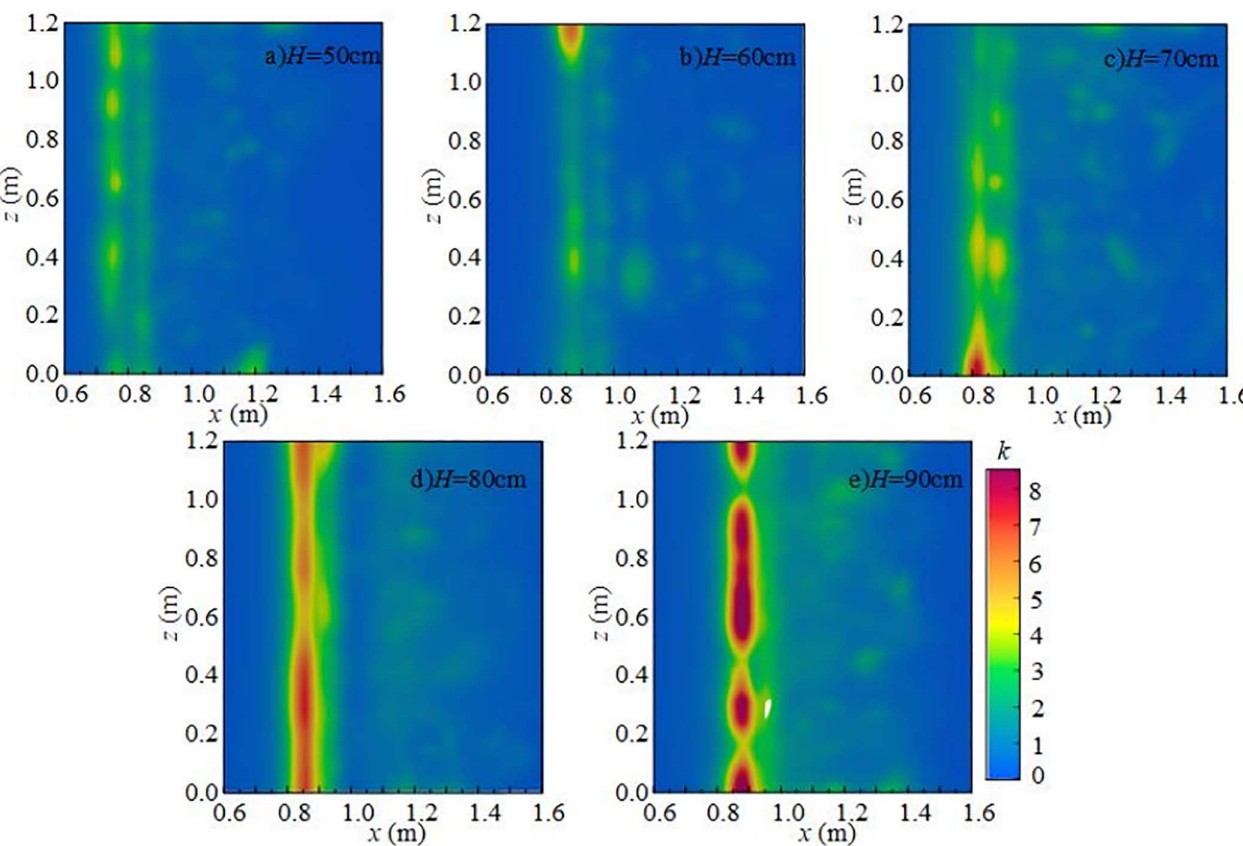

**Fig 22. Spanwise variation of turbulent kinetic energy with different wide-crested weir heights under $Q=75$ L/s and $h=10$ cm.**

where $\overline{S}_{ij} = (u_{ij} + u_{ji})/2$ is the symmetric component of the velocity gradient tensor (i.e., the strain rate), and $\overline{\Omega}_{ij} = (u_{ij} - u_{ji})/2$ is the antisymmetric component (i.e., the vorticity magnitude). For discharge flow over a wide-crested weir, the vortex structures are relatively simple, which facilitates computational efficiency and effective vortex identification. When applying the $Q^*$ criterion, an appropriate threshold must typically be selected manually [48]. In this study, regions with $Q^* > 2$ are identified as vortical structures of interest.

**4.3.1 Vortex structure changes with time.** Fig 23 illustrates the temporal evolution of the vortex structure for discharge flow at $Q = 75$L/s, $H = 90$ cm, $h = 50$ cm. The underwater vortex structures are clearly visible, with flow direction discernible through the structure's orientation. Stable vortices persist over time, whereas unstable vortices dissipate rapidly or evolve into turbulent structures. These vortices are commonly associated with flow separation, which leads to irregular motion, strong velocity gradients, and complex turbulent features. Large-scale vortices generated in shear regions undergo continuous stretching, twisting, compression, and fragmentation. This process gradually transfers energy from large-scale vortices to smaller ones, ultimately leading to energy dissipation via viscous effects.

**4.3.2 Effect of downstream static water depth on vortex structure.** Fig 24 presents the instantaneous vortex structures for discharge flow under different downstream static water depths, with $Q = 75$L/s and $H = 90$ cm held constant. The vortex begins to form just downstream of the weir crest and propagates downstream as the flow progresses. The

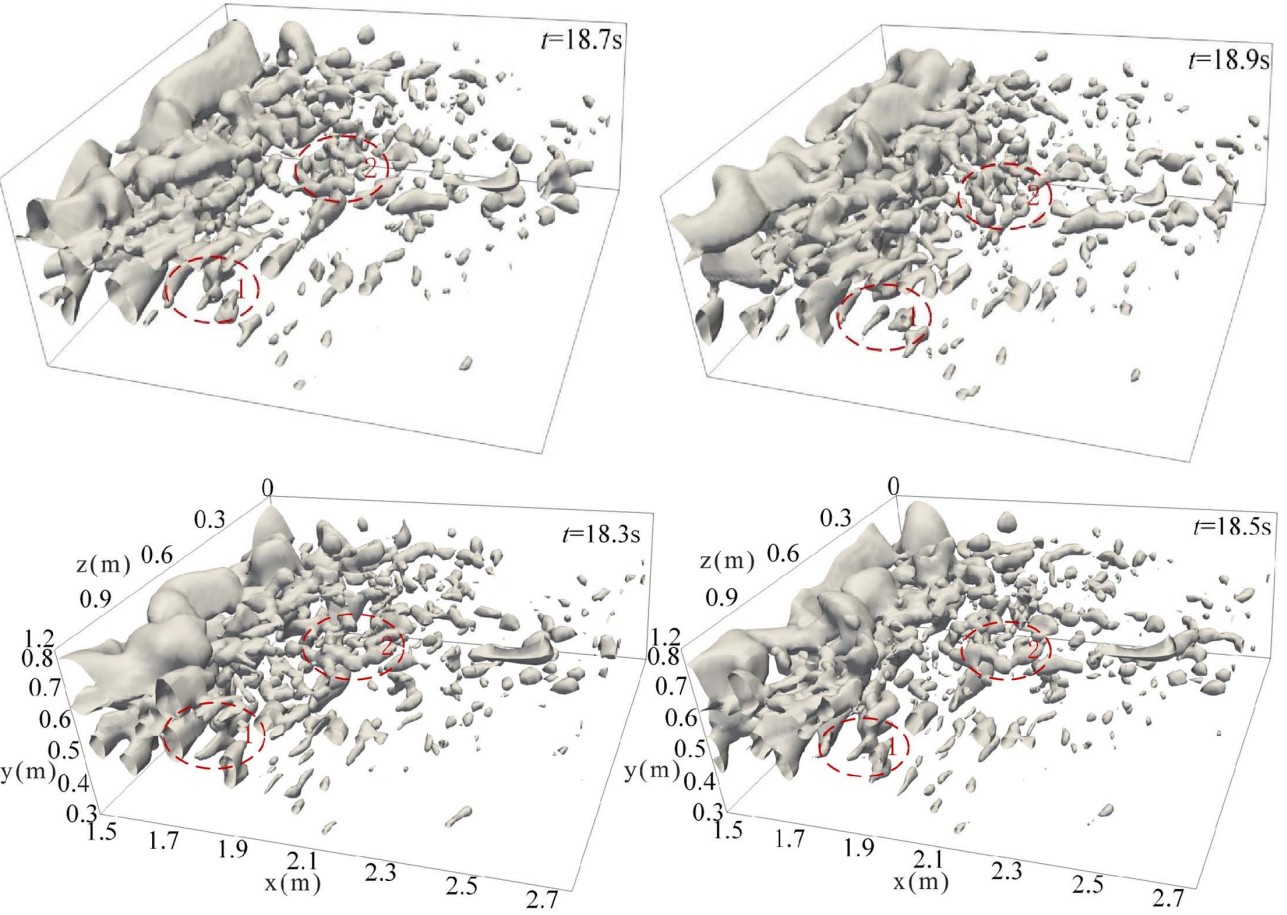

**Fig 23. Specific conditions of discharge flow change with time under $Q = 75$ cm, $H = 90$ cm, $h = 50$ cm.**

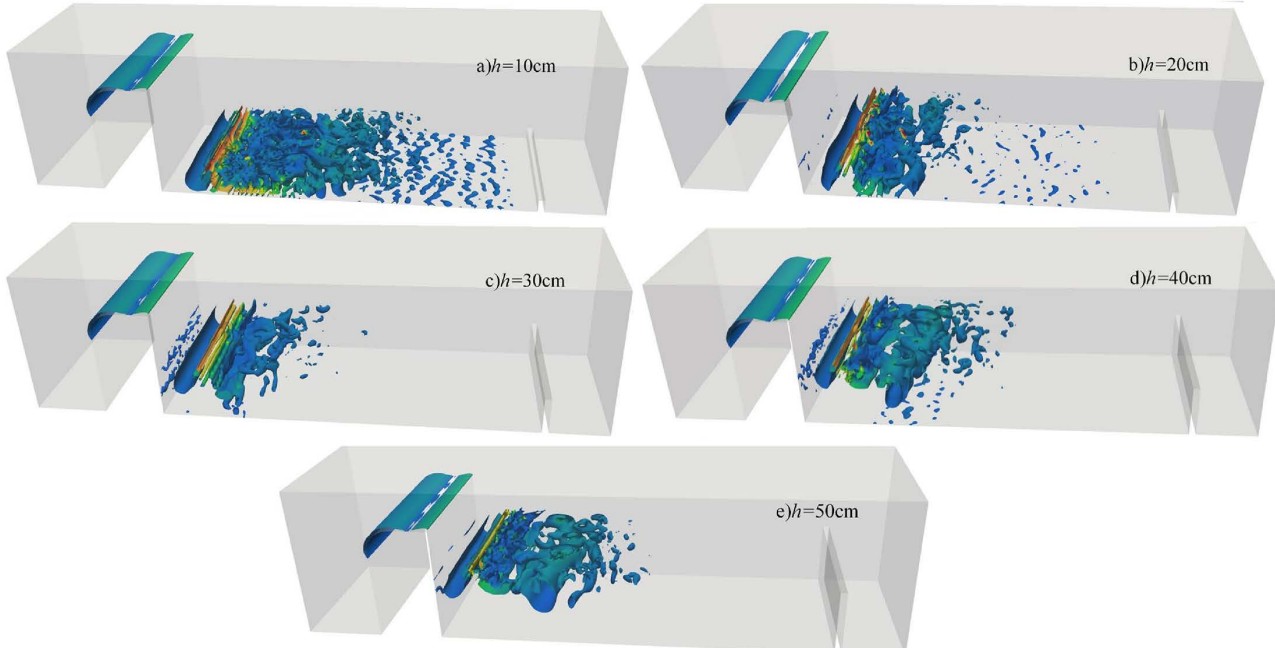

**Fig 24. Transient vortex structure of discharge flow over a wide-crested weir under varying downstream static water depths under $Q=75$L/s and $H=90$ cm.**

color distribution reflects streamwise vorticity: blue and green indicate low-velocity or low-pressure regions, while red and yellow indicate regions of high velocity [48].

When the downstream static water depth is $h=10$ cm, the bottom of the flume displays a dense and complex arrangement of vortices, indicative of strong turbulence. Significant flow separation and disturbance occur at the water–air interface, enhancing entrainment and mixing within the water body and intensifying vortex–fluid interactions. This results in higher levels of underwater noise, particularly due to increased high-frequency noise from bubble collapse.

According to Figs. 24c-24e, as downstream static water depth increases, the flow velocity gradually decreases. This reduction weakens collisions and mixing—both between flow layers and between the flow and the flume bottom—thereby decreasing the number of vortices, dispersing their distribution, and rendering their size more uniform. The overall flow becomes smoother and more stable. Vortex dissipation accelerates, and disturbances at the water–air interface are significantly diminished. Therefore, the hydrodynamic impact on the bottom surface is reduced. The dominant frequency remains in the low-frequency range, associated with turbulence, while the contribution of high-frequency noise from bubble dynamics is substantially reduced due to minimal air–water mixing.

**4.3.3 Effect of incoming flow rate on vortex structure.** Fig 25 shows the instantaneous vortex structures of discharge flow over a wide-crested weir under $h=20$ cm and $H=80$ cm at different incoming flow rates. At low flow rates (Figs 25a–25c), vortex structures are sparse, dispersed, and evenly distributed. Collisions and mixing between fluid layers and with the flume bottom are weak, resulting in relatively low underwater noise. At higher flow rates (Figs 25d–25e), vortex structures become denser and more complex, turbulence intensity increases, and interactions between bubbles and large-scale vortices intensify. Energy levels rise across all frequency bands. Enhanced water–water collisions and air–water mixing increase vortex numbers near the bottom of the flume, leading to a marked rise in underwater noise. Turbulent kinetic energy develops downstream, gradually transitioning into laminar flow, consistent with the coherent structure described in Ref. [47]. During vortex development and evolution, unsteady interactions with the flume bottom

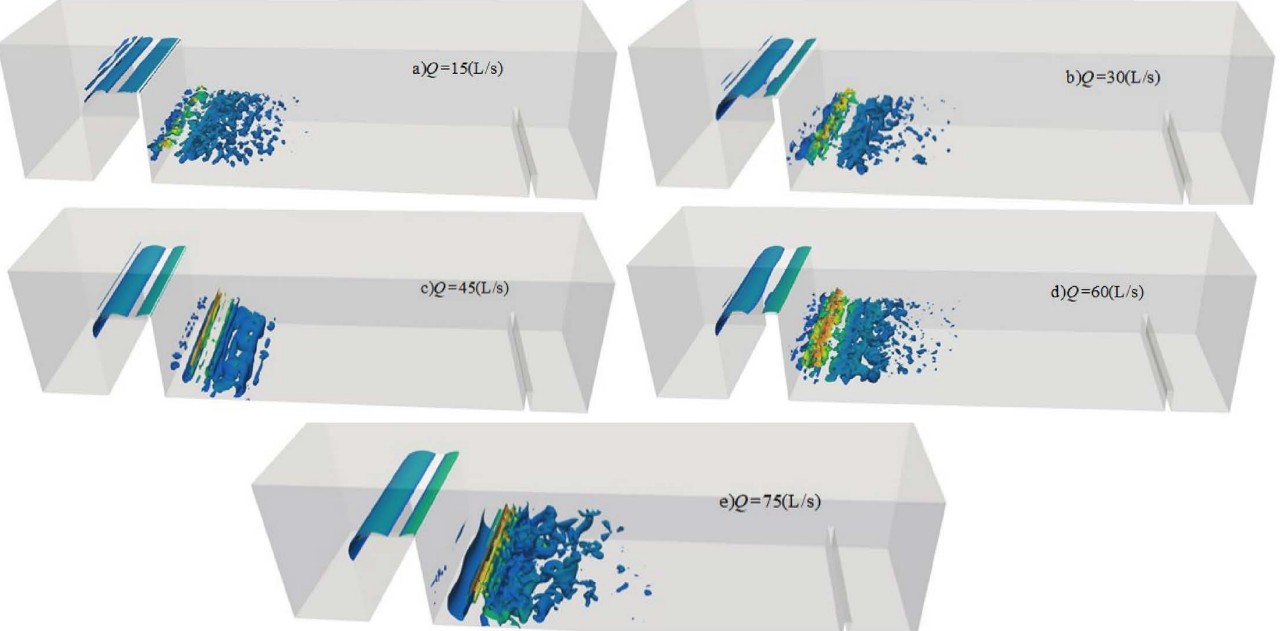

**Fig 25. Instantaneous vortex structure of discharge flow over a wide-crested weir at varying incoming flow rates under $h = 20$ cm and $H = 80$ cm.**

generate strong fluctuating pressures within turbulent regions. Overall, as incoming flow rate increases, vortex structures evolve from sparse to dense, turbulence intensity grows, fluctuating pressure strengthens, and underwater noise increases significantly, with high-frequency components becoming more prominent.

**4.3.4 Effect of wide-crested weir height on vortex structure.** Fig 26 illustrates instantaneous vortex structures of discharge flow under varying wide-crested weir heights at $Q = 75$ L/s and $h = 10$ cm. Variations in weir height exert limited influence on vortex size, which remains relatively uniform. At a weir height of 50 cm, jet impact on the flume bottom is minimal and shear forces are weak, although vortex structures still significantly influence the measuring point. At greater weir heights (Figs 26d–26e), turbulence intensity increases, water–water collisions and air–water mixing intensify, and vortex numbers near the flume bottom rise. This enhances vortex–bottom interactions, producing stronger fluctuating pressures within turbulent regions. Overall, although turbulence intensifies with increasing weir height, the effect on vortex size is limited, and structures remain similar in scale.

Vortices are typically accompanied by flow separation, causing irregular fluid motion, sharp velocity changes, unstable patterns, and strong fluctuations in underwater noise frequency. Large-scale vortices play a critical role in turbulent energy transfer, momentum transport, entrainment, and mixing. Stable vortices may persist, whereas unstable vortices dissipate quickly or transform into turbulence [45]. Energy is transferred from large-scale to small-scale vortices: large vortices carry low-frequency energy, which gradually fragments through nonlinear interactions into smaller vortices corresponding to higher-frequency waves. This cascade represents the transition from low-frequency to high-frequency characteristics, with final dissipation as heat. As shown in Section 3.3, wavelet analysis indicates that low-frequency energy dominates and remains relatively stable, whereas medium- and high-frequency energy decays rapidly. This demonstrates that large-scale vortex structures are the primary source of low-frequency noise, while small-scale vortices contribute to high-frequency noise. When inflow rate is high and downstream static depth is shallow, large-scale vortex generation and evolution are enhanced. Stronger vortex–bottom interactions produce intense fluctuating pressures in turbulent regions, dominated by low-frequency energy characteristics.

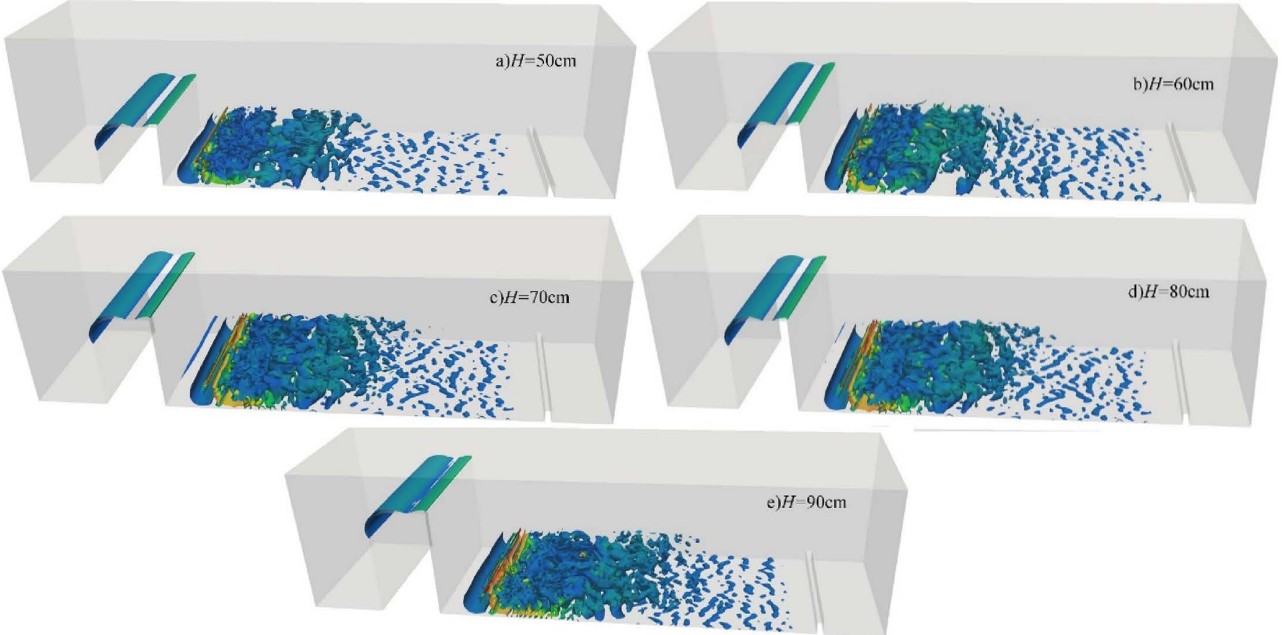

**Fig 26. Transient vortex structure of discharge flow over a wide-crested weir at varying weir heights under *Q*=75L/s, *h*=10cm.**

## 4.4 Effect of fluctuating pressure on underwater noise

To systematically examine the fluctuating pressure characteristics of a wide-crested weir, the effects of weir height, inflow rate, and downstream static water depth on fluctuating pressure at the measuring point were analyzed based on numerical simulation results.

### 4.4.1 Analysis of SPL of fluctuating pressure.

1) Effect of downstream static water depth

Fig 27 shows the relationship between downstream static water depth and fluctuating pressure SPL at the measuring point under different weir heights and inflow conditions. Fluctuating pressure SPL decreases gradually with increasing downstream static water depth, ranging from 140 to 180 dB, with the minimum value occurring at $h=50$ cm. On average, the level decreases by 3.5 dB across different downstream static water depths. Taking $H=90$ cm as an example, the most pronounced reduction occurs at shallow depths ($h=10$ cm~20 cm). When $Q=30$ L/s, an increase in downstream static water depth from 10 cm to 20 cm reduces the SPL by 11.2 dB, whereas an increase from 20 cm to 30 cm reduces it by only 0.2 dB. The variation pattern remains consistent across different weir heights.

2) Effect of incoming flow rate

Fig 28 shows the relationship between incoming flow rate and fluctuating pressure SPL at the measuring point under various weir heights and downstream static depths. SPL increases gradually with flow rate, ranging from 140 dB to 180 dB, with the minimum value at $Q=15$ L/s. The steepest increases occur between $Q=30$ and 45 L/s. On average, the level increases by 2.16 dB across flow rate conditions. For example, when $H=80$ cm and $h=20$ cm, an increase in flow rate from 30 L/s to 45 L/s raises the level by 8.79 dB. In contrast, when $h=40$ cm, an increase from 15 L/s to 30 L/s results in only a 0.29 dB rise.

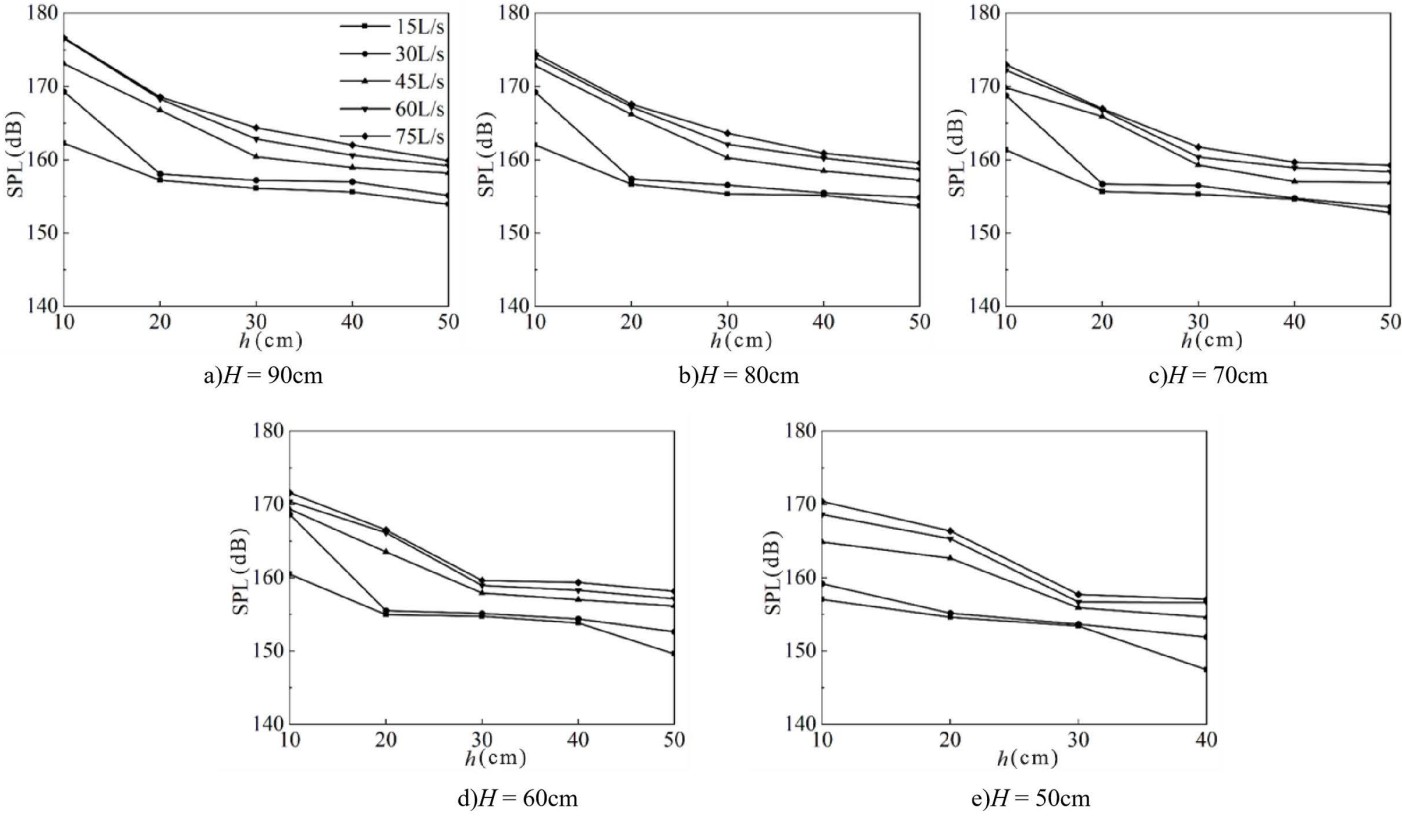

**Fig 27. Relationship between downstream static water depth and fluctuating pressure SPL at the measuring point under varying weir heights and inflow rates.**

3) Effect of weir height

Fig 29 shows the effect of wide-crested weir height on fluctuating pressure SPL at the measuring point under different inflow rates and downstream static depths. Fluctuating pressure SPL increases with weir height, with the minimum value at $H=50$ cm. On average, the level increases by 1.32 dB across the tested heights.

   The trend remains consistent across different downstream static water depths. For example, at $h=40$ cm and $Q=15$ L/s, increasing the weir height from 50 to 60 cm raises the level by 6.30 dB. In contrast, when $Q=75$ L/s, increasing the height from 60 to 70 cm raises the level by only 0.31 dB.

   **4.4.2 Fluctuating pressure RMS variation cloud map.** Fig 30 shows the variation of fluctuating pressure RMS with incoming flow rate under $H=80$ cm and $h=20$ cm, while Fig 31 shows the variation of fluctuating pressure RMS with wide-crested weir height under $Q=75$ L/s and $h=10$ cm. At a low flow rate of 15L/s or a weir height of 50 cm, the discharge enters the downstream region relatively gently, exerting weak impact forces on the flume bottom. Bubble generation is limited, and noise from bubble rupture is relatively small. As inflow and weir height increase, the depth of downstream impact also increases, intensifying the force of the discharge on the flume bottom. The fluctuating pressure RMS gradient of pulsation rises continuously, accompanied by stronger disturbance and tumbling at the water–air interface. This promotes rapid bubble generation and bursting in the downstream region. As turbulence in the flow field intensifies, the fluctuating pressure RMS of fluctuating pressure increases further.

   Fig 32 shows the variation of fluctuating pressure RMS with downstream static water depth at $H=80$ cm and $Q=45$ L/s. When $h=10$ cm, the discharge flow directly impacts the flume bottom, producing strong impact and friction, droplet

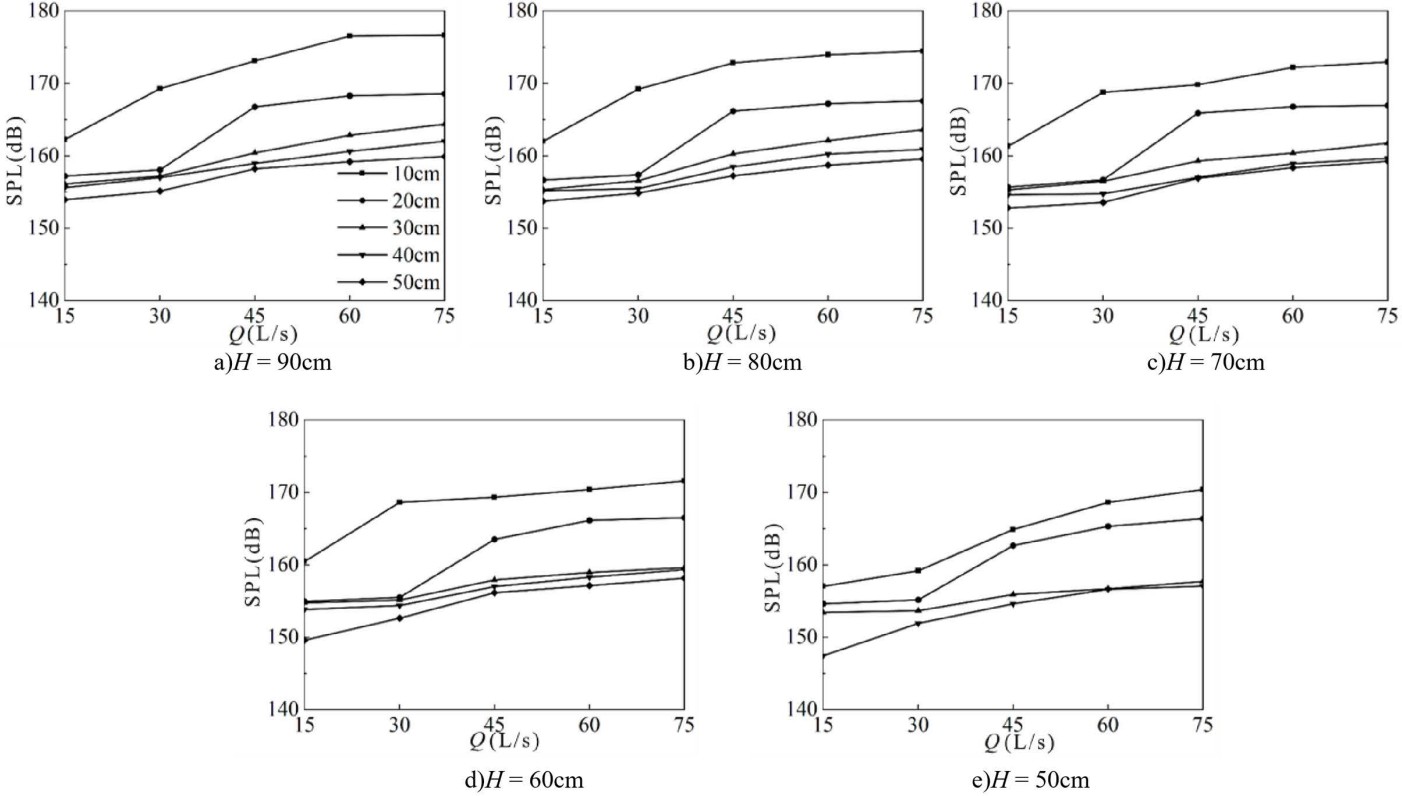

**Fig 28. Relationship between inflow rate and fluctuating pressure SPL at the measuring point under varying weir heights and downstream static water depths.**

splashing, and roll-up rupture at the water–air interface, all of which trigger intense turbulence. With increasing downstream static water depth, impact energy is buffered, transmission and reflection of shock waves are reduced, and vortex formation is suppressed. The fluctuating pressure RMS of fluctuating pressure gradient decreases steadily [44]. Meanwhile, the greater distance between the water–air interface and the measuring point weakens interface disturbance and tumbling, while sound waves from bubble rupture propagate farther and dissipate more, reducing the fluctuating pressure RMS value at the measuring point and lowering the underwater noise SPL.

In summary, increasing inflow and weir height intensifies discharge impacts on the flume bottom and strengthens water–air interface disturbances, thereby raising fluctuating pressure RMS of fluctuating pressure and underwater noise SPL. Conversely, greater downstream static water depth reduces both flow impact and interface disturbance, leading to lower fluctuating pressure RMS values at the measuring point and reduced underwater noise levels.

### 4.4.3 Analysis of frequency characteristics of fluctuating pressure.

1) Effect of wide-crested weir height on fluctuating pressure frequency

Fig 33 shows the time–frequency and frequency–domain diagrams of fluctuating pressure at different weir heights under fixed inflow and downstream static water depth. Under $H=50$ cm, $Q=75$ L/s, and $h=10$ cm, two main frequencies (1.3 Hz and 3.7 Hz) are observed. At $H=60$ cm, $Q=75$ L/s, and $h=30$ cm, the main frequencies shift to 1.3 Hz, 3.7 Hz, and 6.5 Hz, while at $H=70$ cm, $Q=75$ L/s, and $h=40$ cm, the same three frequencies remain dominant. At $H=80$ cm, $Q=75$ L/s, and $h=40$ cm, the main frequencies expand to 0.8 Hz, 3.7 Hz, 6.5 Hz, and 17.2 Hz. The spectra at $H=90$ cm are similar

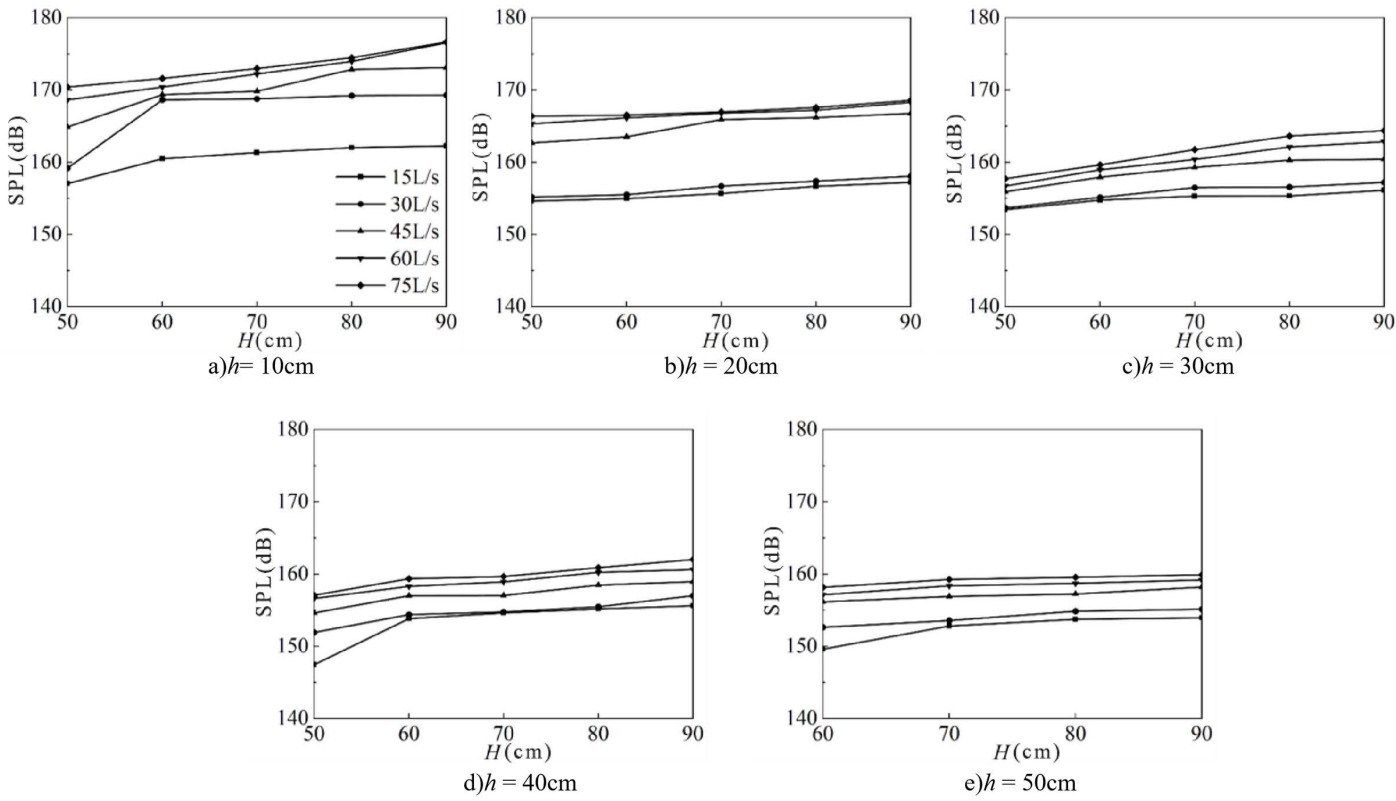

**Fig 29. Relationship between wide-crested weir height and fluctuating pressure SPL at the measuring point under varying inflow rates and downstream static water depths.**

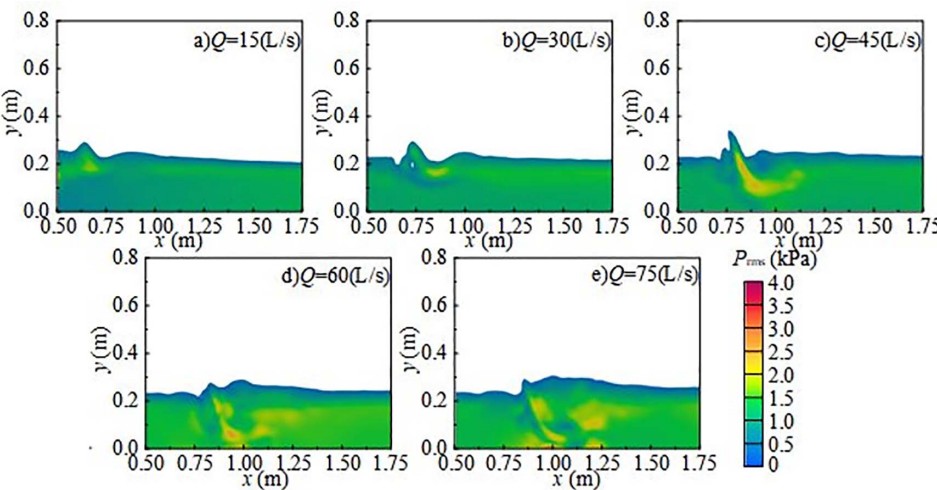

**Fig 30. Cloud map of fluctuating pressure RMS variation with incoming flow rate under $H=80\,cm$ and $h=20\,cm$.**

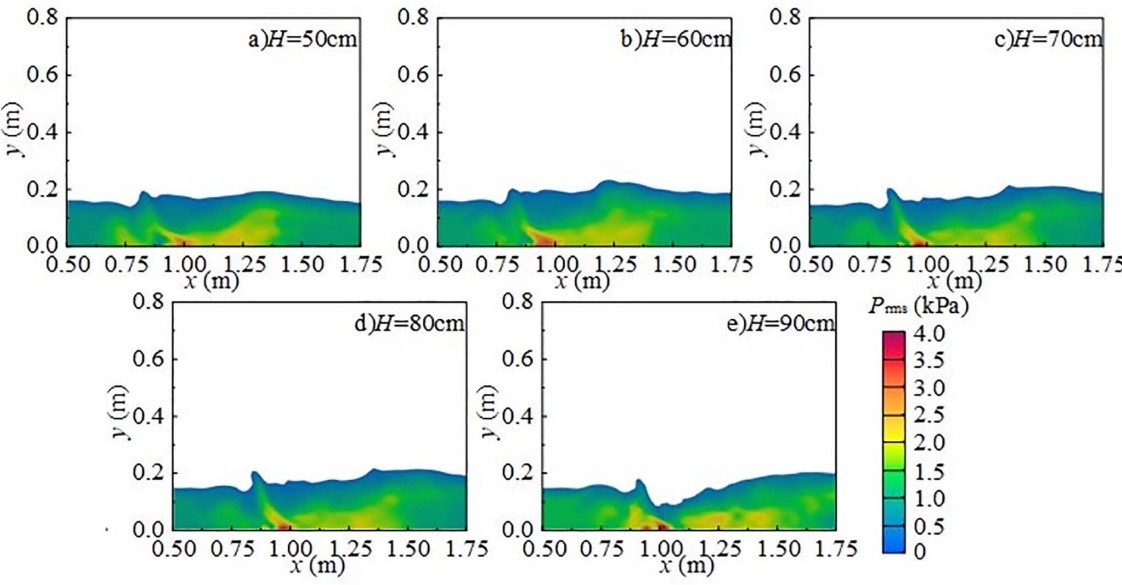

**Fig 31. Cloud map of fluctuating pressure RMS variation with wide-crested weir height under $Q=75$ L/s and $h=10$ cm.**

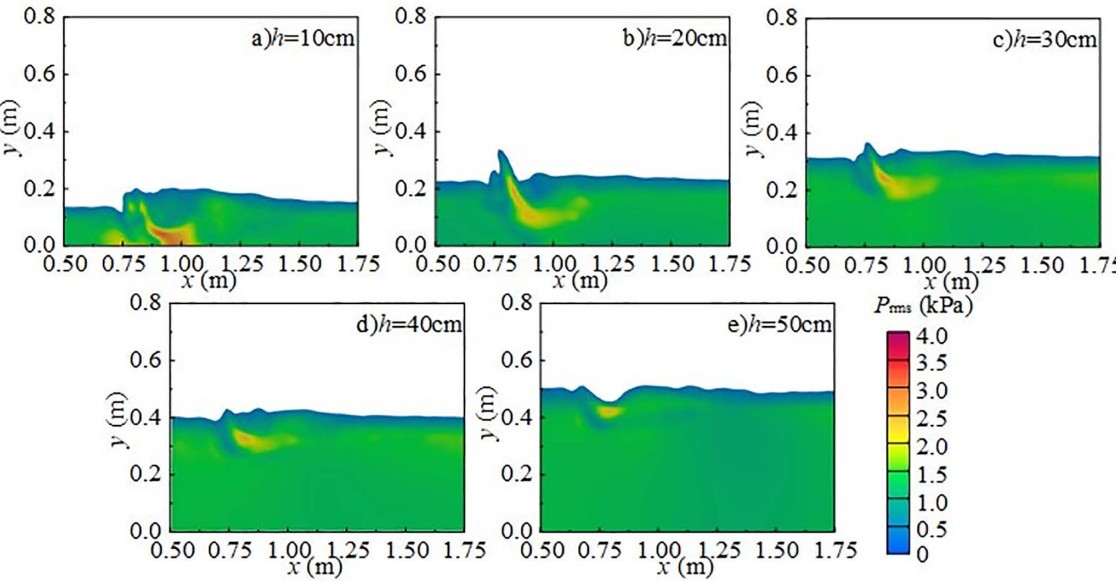

**Fig 32. Variation of fluctuating pressure RMS with downstream static water depth under $H=80$ cm and $Q=45$ L/s.**

to those at $H=80$ cm. Although the frequencies vary under different operating conditions, the dominant ones consistently concentrate around 0.2 Hz, 0.8 Hz, 1.3 Hz, 1.5 Hz, 3.2 Hz, and 6.2 Hz.

The amplitude decays rapidly as frequency increases, indicating that fluctuating pressure energy is concentrated in the low-frequency band (0~20 Hz). With increasing weir height, the amplitude declines relatively, as the higher discharge

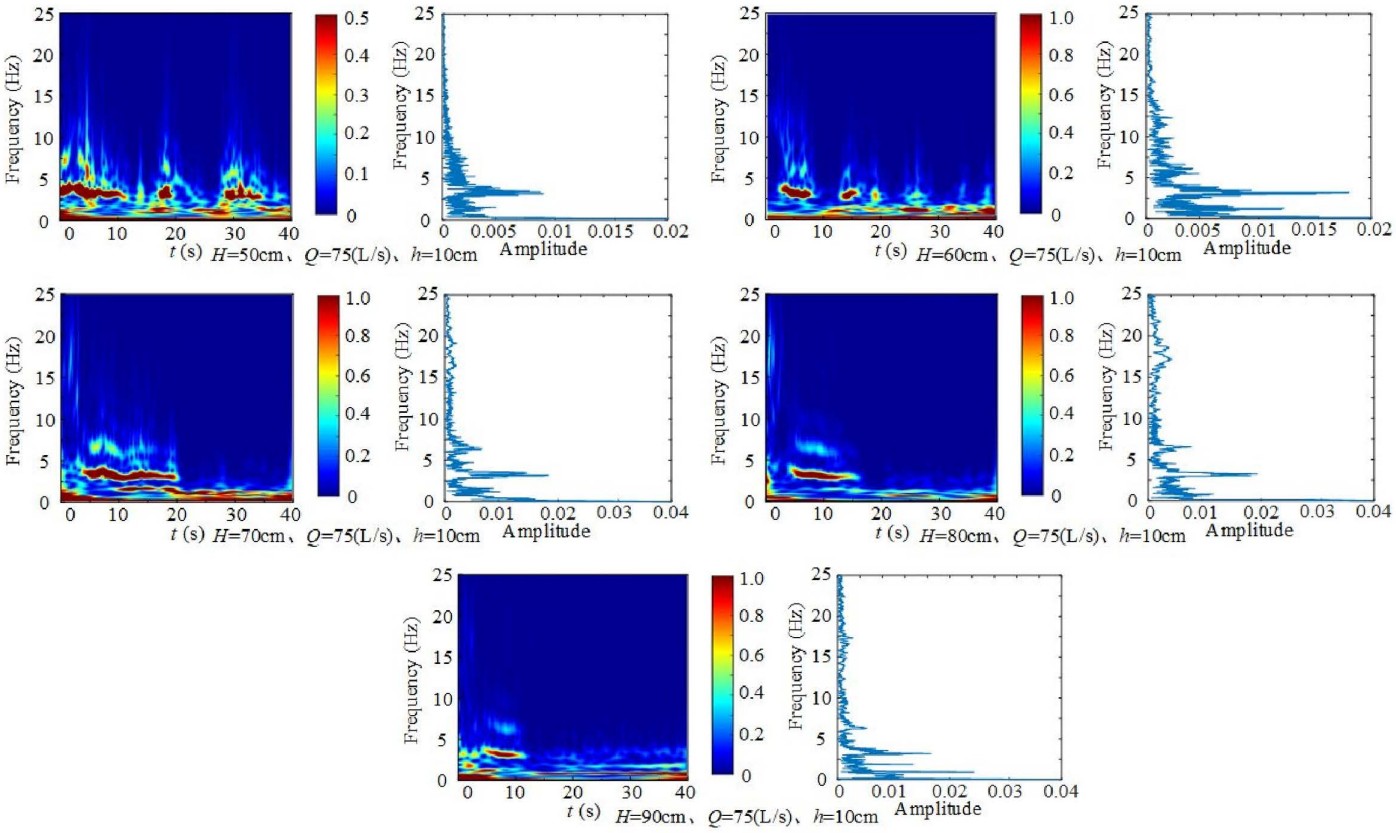

**Fig 33. Time–frequency and frequency-domain diagrams of fluctuating pressure at different weir heights under specified inflow and downstream static water depths.**

head enhances the kinetic energy of the jet impacting the downstream surface, which intensifies turbulence development and increases pressure fluctuation energy at the measurement point. Nevertheless, the dominant frequencies remain stable at approximately 0.8 Hz, 3 Hz, and 6.5 Hz, showing that height adjustment influences energy intensity rather than frequency position.

In summary, the 0~20 Hz range is the main energy contribution band. Although operating conditions cause some variation, the dominant frequencies consistently cluster near 0.8 Hz, 3 Hz, and 6.5 Hz.

2)   Effect of downstream static water depth on fluctuating pressure frequency

Fig 34 illustrates the influence of downstream static water depth under fixed inflow and weir height. At $H = 90$ cm, $Q = 75$ L/s and $h = 10$ cm10 cm, the dominant frequencies are 1.3 Hz, 2.1 Hz, 3.7 Hz and 6.5 Hz, respectively. When $H = 90$ cm, $Q = 75$ L/s and $h = 20$ cm, they reduce to 0.8 Hz, 1.3 Hz, and 3.2 Hz. When $H = 90$ cm, $Q = 75$ L/s, $h = 30$ cm, to 0.8 Hz, 1.5 Hz, and 3.2 Hz. When $H = 90$ cm, $Q = 75$ L/s, $h = 40~50$ cm, the sole dominant frequency is 0.8 Hz. Although frequencies vary with depth, most are concentrated at 0.8 Hz, 1.3 Hz, 3.2 Hz, and 6.5 Hz.

The amplitude decays rapidly with frequency, and higher-frequency signals dissipate faster. With greater downstream static water depth, the amplitude decreases because the jet's kinetic energy dissipates before reaching the channel bed, which slows turbulence development.

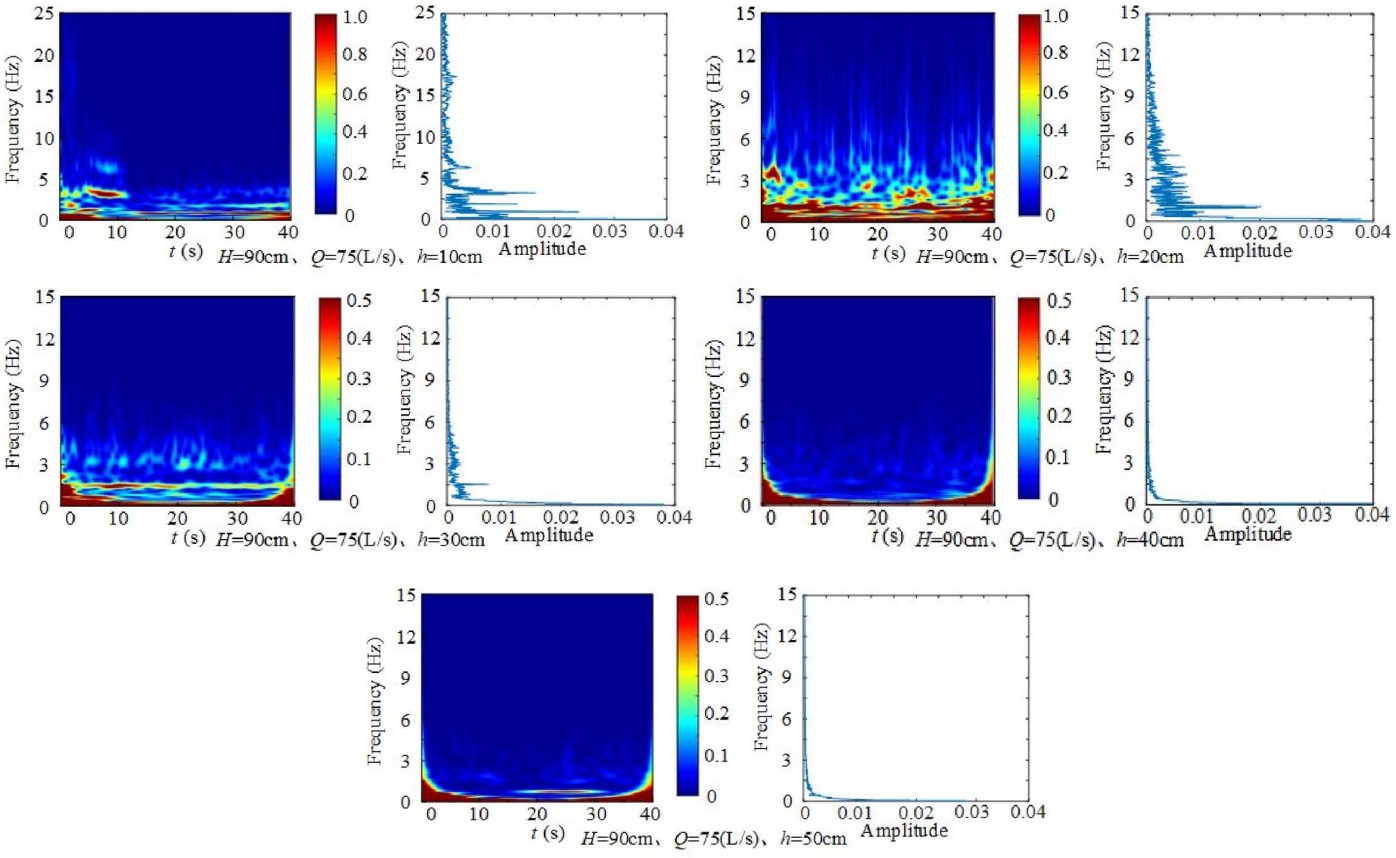

**Fig 34. Time–frequency and frequency-domain diagrams of fluctuating pressure with varying downstream static water depth under fixed inflow and weir height.**

3) Effect of inflow rate on fluctuating pressure frequency

Fig 35 presents time–frequency diagrams of fluctuating pressure at different inflow rates under constant weir height and downstream static water depth. At $H=90$ cm and $h=10$ cm, dominant frequencies occur at 0~0.2 Hz, 1.3 Hz, 2.1 Hz, 3.2 Hz and 6.2 Hz. At $h=20$ cm, four dominant frequencies are observed at 0~0.2 Hz, 0.8 Hz, 1.3 Hz and 3.2 Hz. Although frequency components vary slightly among groups, they remain concentrated in the 0–6 Hz range. Amplitude decreases sharply with frequency, but overall amplitude rises with inflow due to increased kinetic energy transfer to the downstream region and stronger flow impacts.

In conclusion, increasing inflow and weir height amplifies fluctuating pressure at dominant frequencies, as stronger impacts intensify turbulent kinetic energy and vortex stretching, compression, twisting, and splitting, which in turn strengthen fluctuating pressure at the measuring point. In contrast, greater downstream static water depth decreases fluctuating pressure amplitude by buffering the discharge flow and reducing its direct impact, thereby suppressing turbulent kinetic energy and vortex development. Although amplitudes vary, dominant frequency components remain concentrated around 0.8 Hz, 1.6 Hz, 3 Hz, and 6.5 Hz. Thus, variations are mainly reflected in amplitude rather than frequency distribution. Compared with weir height, downstream static water depth and inflow rate exert stronger effects on frequency characteristics, not only increasing overall frequency energy but also amplifying peaks at higher frequencies. These findings provide theoretical and technical insights into the flow dynamics of wide-crested weirs and their influence on downstream hydraulics.

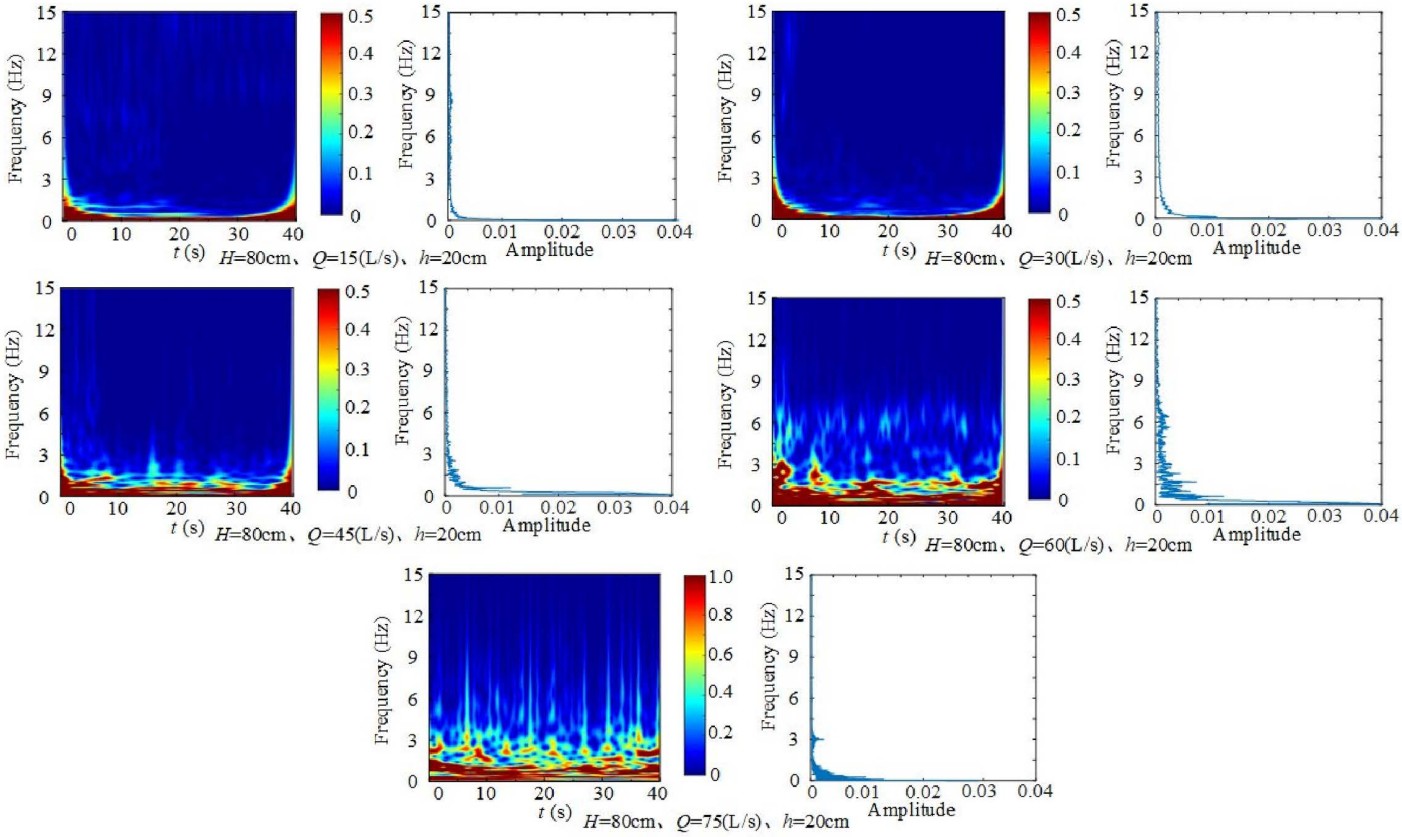

**Fig 35. Time–frequency and frequency-domain diagrams of fluctuating pressure with varying inflow under fixed weir height and downstream static water depth.**

## 5 Conclusion

This study investigated the characteristics and mechanisms of underwater noise generated by discharge flow from wide-crested weirs through physical experiments and numerical simulations. The effects of inflow rate, downstream static water depth, and weir height on underwater noise SPL and fluctuating pressure were analyzed experimentally, while the mechanisms of noise formation from bubble–water mixed flow with a free surface were explored numerically. The main conclusions are as follows:

1) The SPLs of underwater noise and fluctuating pressure increase with inflow rate and weir height, and decrease with downstream static water depth. Their relative influence is ranked as: downstream static water depth＞inflow rate＞weir height. Fluctuating pressure SPL is strongly and positively correlated with underwater noise SPL, enabling prediction of underwater noise using the following relationship: $L_{pw} = 1.23\text{SPL} - 61.4$

2) The dominant frequency of underwater noise lies in the low-frequency range, with relatively high energy between 0～20 Hz and 200～300 Hz. Increasing inflow or weir height, or decreasing downstream static water depth, enhances contributions from intermediate frequencies. Stronger signal features emerge in the 400～600 Hz band, but energy decays rapidly. With greater downstream static water depth, RMS of fluctuating pressure at the measuring point decreases and underwater noise weakens. Conversely, higher inflow and greater weir height intensify RMS of fluctuating pressure and increase underwater noise levels.

3) Numerical simulation results show that when the main flow tongue enters the downstream water body, the impact on the fluid, vortex structural changes, and bubble rupture at the water–air interface collectively influence fluctuating pressure at the measuring point. Increasing downstream static water depth reduces fluctuating pressure and underwater noise SPLs. Fluctuating pressure energy is concentrated in the low-frequency range of 0~6 Hz, with stable dominant frequencies around 0.8, 1.6, and 3 Hz. Higher inflow rates, greater weir height, and lower downstream static water depths promote the generation of mid- and high-frequency noise through bottom impacts and small-scale vortices, though their energy dissipates rapidly.

In summary, the findings provide theoretical guidance for the design and noise control of wide-crested weirs. Measures such as increasing downstream static water depth in the flow impact region, designing smoother transition sections, and mitigating abrupt velocity changes and turbulence can reduce pressure fluctuations and achieve effective noise suppression.

## Supporting information

**S1 Fig. The frequency spectra at different downstream static water depths and inflow rates, with weir height $H=50$ cm obtained using the FFT.**
(TIF)

**S2 Fig. The frequency spectra at different downstream static water depths and inflow rates, with weir height $H=60$ cm obtained using the FFT.**
(TIF)

**S3 Fig. The frequency spectra at different downstream static water depths and inflow rates, with weir height $H=70$ cm obtained using the FFT.**
(TIF)

**S4 Fig. The frequency spectra at different downstream static water depths and inflow rates, with weir height $H=80$ cm obtained using the FFT.**
(TIF)

**S5 Fig. The frequency spectra at different downstream static water depths and inflow rates, with weir height $H=90$ cm obtained using the FFT.**
(TIF)

## Acknowledgments

The authors gratefully acknowledge the teachers at Changsha University of Science and Technology for their valuable guidance and assistance in the experimental process.

## Author contributions

**Conceptualization:** Qingxiang Shui.

**Data curation:** Yuxin Chen, Yi Fan, Yang Dai.

**Formal analysis:** Yuxin Chen.

**Funding acquisition:** Qingxiang Shui, Daguo Wang.

**Investigation:** Qingxiang Shui.

**Methodology:** Qingxiang Shui, Daguo Wang, Tao Yu.

**Software:** Qingxiang Shui.

**Validation:** Qingxiang Shui, Tao Yu.

**Writing – original draft:** Qingxiang Shui.

**Writing – review & editing:** Qingxiang Shui.

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
