## [Decision Letter · Decision Letter 0]

9 Dec 2025

Dear Dr. Wang,

Thank you for submitting your manuscript to PLOS ONE. After careful consideration, we feel that it has merit but does not fully meet PLOS ONE’s publication criteria as it currently stands. Therefore, we invite you to submit a revised version of the manuscript that addresses the points raised during the review process.

We look forward to receiving your revised manuscript.

Kind regards,

Shicheng Li

Academic Editor

PLOS One

“This study was supported by the National Natural Science Foundation of China (Grant Number 42171108), the Sichuan International Science and Technology Innovation Cooperation Project (Grant Number 2025YFHZ0223) and the Open Research Subject of Key Laboratory of Fluid and Power Machinery (Xihua University), Ministry of Education (Grant Number LTDL-2025014).”

6. PLOS requires an ORCID iD for the corresponding author in Editorial Manager on papers submitted after December 6th, 2016. Please ensure that you have an ORCID iD and that it is validated in Editorial Manager. To do this, go to ‘Update my Information’ (in the upper left-hand corner of the main menu), and click on the Fetch/Validate link next to the ORCID field. This will take you to the ORCID site and allow you to create a new iD or authenticate a pre-existing iD in Editorial Manager.

Reviewers' comments:

Reviewer's Responses to Questions

**Comments to the Author**

1. Is the manuscript technically sound, and do the data support the conclusions?

Reviewer #1: Yes

Reviewer #2: Yes

2. Has the statistical analysis been performed appropriately and rigorously?

Reviewer #1: I Don't Know

Reviewer #2: Yes

3. Have the authors made all data underlying the findings in their manuscript fully available?

Reviewer #1: No

Reviewer #2: Yes

4. Is the manuscript presented in an intelligible fashion and written in standard English?

Reviewer #1: Yes

Reviewer #2: Yes

Reviewer #1: This works integrates physical experiments and numerical simulations to provide theoretical guidance and technical support for mitigating underwater noise generated by discharge flow from wide-crested weirs.

Although the initial intent of the authors seems valuable, the reviewer has detected flaws in the presentation and analysis of the experimental results as well as in the grid convergence process impacting numerical results that prevents its publication in PLOS ONE in the present form

Reviewer #2: This manuscript is about underwater noise characteristics and mechanism of discharge flow from wide-crested weir and it provides theoretical guidance and technical support for mitigating underwater noise generated by discharge flow from wide-crested weirs. The manuscript topic is within the scope of the journal. The manuscript is well-written and provides a comprehensive study for the noise generated by weir discharging. Please see the attached file for more details.

**Do you want your identity to be public for this peer review?** For information about this choice, including consent withdrawal, please see our Privacy Policy

Reviewer #1: No

Reviewer #2: No

---

## [Author Response · Author response to Decision Letter 1]

22 Jan 2026

The comments raised by the reviewers have been addressed in detail in the “Response to Reviewers” file included in the Attach files section. The revisions suggested by the editor have also been implemented in accordance with the relevant requirements. Should any further modifications be required, we would appreciate it if the editor could kindly inform us.

The following is particularly worth noting.

Journal requirements-3. We note that the grant information you provided in the ‘Funding Information’ and ‘Financial Disclosure’ sections do not match.When you resubmit, please ensure that you provide the correct grant numbers for the awards you received for your study in the ‘Funding Information’ section.

---

## [Decision Letter · Decision Letter 1]

23 Feb 2026

Study on underwater noise characteristics and mechanism of discharge flow from wide-crested weir

PONE-D-25-48325R1

Dear Dr. Wang,

We’re pleased to inform you that your manuscript has been judged scientifically suitable for publication and will be formally accepted for publication once it meets all outstanding technical requirements.

Kind regards,

Shicheng Li

Academic Editor

PLOS One

Additional Editor Comments (optional):

Reviewers' comments:

Reviewer's Responses to Questions

**Comments to the Author**

Reviewer #1: All comments have been addressed

Reviewer #2: All comments have been addressed

2. Is the manuscript technically sound, and do the data support the conclusions?

Reviewer #1: Yes

Reviewer #2: Yes

3. Has the statistical analysis been performed appropriately and rigorously?

Reviewer #1: Yes

Reviewer #2: Yes

4. Have the authors made all data underlying the findings in their manuscript fully available?

Reviewer #1: Yes

Reviewer #2: Yes

5. Is the manuscript presented in an intelligible fashion and written in standard English?

Reviewer #1: Yes

Reviewer #2: Yes

Reviewer #1: The reviewer thanks the authors for the good improvements they made to their paper.

It is now well enough written to be published should the editor decide so.

Reviewer #2: I think the manuscript is now a well-written manuscript and authors were able to address all comments professionally.

**Do you want your identity to be public for this peer review?** For information about this choice, including consent withdrawal, please see our Privacy Policy

Reviewer #1: No

Reviewer #2: No

---

## [Editor Report · Acceptance letter]

PONE-D-25-48325R1

PLOS One

Dear Dr. Wang,

I'm pleased to inform you that your manuscript has been deemed suitable for publication in PLOS One. Congratulations! Your manuscript is now being handed over to our production team.

Kind regards,

on behalf of

Dr. Shicheng Li

Academic Editor

PLOS One